

# Evaluating the effect of alternative carbon allocation schemes in a land surface model (CLM4.5) on carbon fluxes, pools and turnover in temperate forests

Francesc Montané[1], Andrew M. Fox[1], Avelino F. Arellano[2], Natasha MacBean[1], M. Ross Alexander[1,3], Alex Dye[4], Daniel A. Bishop[5], Valerie Trouet[3], Flurin Babst[6,7], Amy E. Hessl[4], Neil Pederson[8], Peter D. Blanken[9], Gil Bohrer[10], Christopher M. Gough[11], Marcy E. Litvak[12], Kimberly A. Novick[13], Richard P. Phillips[14], Jeffrey D. Wood[15], David J.P. Moore[1]

[1]School of Natural Resources and the Environment, University of Arizona, Tucson, Arizona, 85721, USA
[2]Department of Hydrology and Atmospheric Sciences, University of Arizona, Tucson, Arizona, 85721, USA
[3]Laboratory of Tree-Ring Research, University of Arizona, Tucson, Arizona, 85721, USA
[4]Department of Geology and Geography, West Virginia University, Morgantown, West Virginia, 26506, USA
[5]Division of Biology and Paleo Environment, Lamont-Doherty Earth Observatory, Columbia University, Palisades, New York, 10964, USA
[6] Dendro Sciences Unit, Swiss Federal Research Institute WSL, Zürcherstrasse 111, CH-8903 Birmensdorf, Switzerland
[7]W. Szafer Institute of Botany, Polish Academy of Sciences, ul. Lubicz 46, 31-512 Krakow, Poland
[8]Harvard Forest, Harvard University, Petersham, Massachusetts, 01366, USA
[9]Department of Geography, University of Colorado, Boulder, Colorado, 80309, USA
[10]Department of Civil, Environmental, and Geodetic Engineering, The Ohio State University, Columbus, Ohio, 43210, USA
[11]Department of Biology, Virginia Commonwealth University, Richmond, Virginia, 23284, USA
[12]Department of Biology, University of New Mexico, Albuquerque, New Mexico, 87131, USA
[13]School of Public and Environmental Affairs, Indiana University, Bloomington, Indiana, 47405, USA
[14]Department of Biology, Indiana University, Bloomington, Indiana, 47405, USA
[15]School of Natural Resources, University of Missouri, Columbia, Missouri, 65211, USA

*Correspondence to*: Francesc Montané (francesc.montane@gmail.com)

**Abstract.** How carbon (C) is allocated to different plant tissues (leaves, stem and roots) determines C residence time and thus remains a central challenge for understanding the global C cycle. We used a diverse set of observations (AmeriFlux eddy covariance tower observations, biomass estimates from tree-ring data, and Leaf Area Index (LAI) measurements) to compare C fluxes, pools, and LAI data with those predicted by a Land Surface Model (LSM), the Community Land Model (CLM4.5). We ran CLM for nine temperate (including evergreen and deciduous) forests in North America between 1980 and 2013 using four different C allocation schemes: i) Dynamic C allocation scheme (named "D-CLM") with one dynamic allometric parameter, which allocates C to the stem and leaves to vary in time as a function of annual Net Primary Production (NPP). ii) An alternative dynamic C allocation scheme (named "D-Litton"), where, similar to (i) C allocation is a dynamic function of annual NPP, but unlike (i) includes two dynamic allometric parameters involving allocation to leaves, stem and coarse roots iii-iv) Two fixed C allocation schemes, one representative of observations in evergreen (named "F-Evergreen") and the other of observations in deciduous forests (named "F-Deciduous"). D-CLM generally overestimated Gross Primary Production (GPP) and ecosystem respiration, and underestimated Net Ecosystem Exchange (NEE). In D-CLM, initial aboveground





biomass in 1980 was largely overestimated (between 10527 and 12897 gCm$^{-2}$) for deciduous forests, whereas aboveground biomass accumulation through time (between 1980 and 2011) was highly underestimated (between 1222 and 7557 gCm$^{-2}$) for both evergreen and deciduous sites due to a lower stem turnover rate in the sites than the one used in the model. D-CLM overestimated LAI in both evergreen and deciduous sites because the leaf C-LAI relationship in the model did not match the observed leaf C-LAI relationship at our sites. Although the four C allocation schemes gave similar results for aggregated C fluxes, they translated to important differences in long-term aboveground biomass accumulation and aboveground NPP. For deciduous forests, D-Litton gave more realistic $C_{stem}/C_{leaf}$ ratios and strongly reduced the overestimation of initial aboveground biomass, and aboveground NPP for deciduous forests by D-CLM. We identified key structural and parameterization deficits that need refinement to improve the accuracy of LSMs in the near future. That could be done by addressing some of the current model assumptions about C allocation and the associated parameter uncertainty.

Our results highlight the importance of using aboveground biomass data to evaluate and constrain the C allocation scheme in the model, and in particular, the sensitivity to the stem turnover rate. Revising these will be critical to improving long-term C processes in LSMs, and improve their projections of biomass accumulation in forests.

# 1 Introduction

Over the last half century, on average a little more than a quarter of global $CO_2$ emissions were absorbed by terrestrial carbon (C) sinks (Le Quéré et al., 2015), with forests accounting for most (Malhi et al., 2002; Bonan, 2008; Pan et al., 2011; Baldocchi et al., 2016). The interannual variability in the land C sink is high, representing up to 80% of annual $CO_2$ emissions (Le Quéré et al., 2009). The mechanism by which forests accumulate C is through photosynthetic uptake and allocation of the C to biomass in different plant pools (leaf, stem and root). The C stored in biomass stocks are determined mainly by the C fluxes and the C allocation amongst plant pools.

Recent modeling studies have shown that simultaneous consideration of C allocation and residence times is crucial to better understand their combined effects on biomass accumulation (Bloom et al., 2016; Koven et al., 2015; De Kauwe et al., 2014). Carbon residence time in the different plant pools (leaf, stem, and root) influences whether ecosystems are projected to act as C sources or sinks (Delbart et al., 2010; Friend et al., 2014). Once C is taken up by the plant, the carbon is allocated either to short-lived leaf or fine-root tissues, or to longer lived woody tissues. Plants that allocate a greater proportion of C to tissues with long residence times (e.g. stem) have a higher standing biomass than the plants that allocate a greater proportion of C to tissues with short residence times (e.g. leaf). Ecological theory suggests that variation in C allocation to different plant pools is governed by functional trade-offs (Tilman, 1988); with plants investing in either aboveground or belowground tissues depending on which strategy would maximise growth and reproduction. If the functional trade-off hypothesis is relevant at forest or regional scales, land surface models (LSMs) for forests should represent it using dynamic C-allocation schemes, which are responsive to above (e.g. light) and belowground (e.g. water or nutrients) factors that limit growth.



Currently, many LSMs use a simplistic approach of allocating C between pools using fixed ratios (e.g. fixed coefficient models) (De Kauwe et al., 2014) for each Plant Functional Type (PFT), assuming that allocation fractions are not affected by environmental conditions. A global syntheses of evergreen and deciduous forests show differences in inferred C allocation patterns, for example, the percentage of NPP allocated to leaves that is greater in deciduous than in evergreen forests (Luyssaert

et al., 2007), though many LSMs use the same fractional allocation for both of these forest types. LSMs poorly represent observed relationships between productivity and different pools of biomass within tropical forests (Delbart et al., 2010; Malhi et al., 2011; Negron-Juarez et al., 2015). Eddy covariance observations are commonly used to parameterize and benchmark LSMs either at single sites or, using geospatial scaling methods, across regions or the globe (Baldocchi et al., 2001; Friend et al., 2007; Randerson et al., 2009; Zaehle and Friend 2010; Mahecha et al., 2010; Bonan et al., 2011). Biosphere-atmosphere

fluxes indicate the balance between the amount of $CO_2$ entering the system through assimilation (e.g. photosynthesis) and the amount of $CO_2$ leaving the system through respiration (e.g. ecosystem respiration) but do not provide information on allocation between pools (Richardson et al., 2010). Studies that focus on C allocation to the different plant pools are not common (e.g. Gower et al., 2001; Franklin et al., 2012). It is difficult to measure allocation to different pools at ecosystem or landscape scales and instead we infer what partitioning was required to result in different biomass pools. Some forest inventory data

includes estimates of the average biomass within the leaf, wood and root pool, and these can be used to parameterize and benchmark models (Caspersen et al., 2000; Brown, 2002; Houghton, 2005; Keith et al., 2009). Studies focusing simultaneously on C pools, fluxes and allocation are relatively rare (Wolf et al., 2011; Xia et al., 2015; Bloom et al., 2016; Thum et al., 2017), in part because collecting biometric data in addition to flux data is very labour intensive.

In this study, we evaluate mechanisms by which C is stored over multiple decades in plant biomass using corresponding eddy covariance flux towers and biometric measurements of C storage in different pools. We collated biometric data, where available, for AmeriFlux sites and supplemented these data with novel aboveground biomass estimates from tree-ring data for AmeriFlux sites (Alexander et al., in review). We evaluate two dynamic C allocation schemes (Oleson et al., 2013; Litton et al., 2007) and two fixed C allocation schemes (Luyssaert et al., 2007) within the Community Land Model (CLM) against C

fluxes, stocks, and Leaf Area Index (LAI) data at nine temperate North American forest ecosystems.

## 2 Methods

### 2.1 Study sites

Nine temperate forests widely distributed throughout the USA were selected for this study, including four evergreen (Niwot Ridge, Valles Caldera Mixed Conifer, Howland Forest, and Duke Forest Loblolly Pine) and five deciduous forests (University

of Michigan Biological Station, Missouri Ozark, Harvard Forest, Morgan Monroe State Forest, and Duke Forest Hardwoods) (Table 1). All the selected forests are AmeriFlux sites (http://ameriflux.lbl.gov/), a network of eddy covariance sites measuring



ecosystem C, water, and the energy fluxes in North and South America. AmeriFlux datasets provide central connections between terrestrial ecosystem processes and climate responses from site to continental scale, and are part of FLUXNET, a global network of eddy covariance measurements being made on all continents.

**2.2 Observations**

We compiled different data streams from diverse sources for the sites (Table 1) for benchmarking C fluxes, C pools, and LAI in the model experiments. Some of the data were only available for a subset of sites and years (Table 1).

Eddy covariance tower data were derived from the AmeriFlux L2 gap-filled data product for all sites (Table 1), except for Niwot Ridge where only the AmeriFlux L2 with-gaps data product was available and there we used the REddyProc package (Reichstein et al., 2005) to gap-fill and partition the data. Half-hourly eddy covariance flux data were aggregated to annual values at all sites. While partitioning and uncertainty analysis were available from the FLUXNET2015 dataset (http://fluxnet.fluxdata.org/data/fluxnet2015-dataset/) only for some sites, but not for all, we opted to use only AmeriFlux L2 data and process all sites using the same protocol.

Aboveground biomass between 1980 and 2011 was estimated for all sites (Table 1) using a dendrochronological sampling technique (Dye et al., 2016; Alexander et al., in review), to reconstruct year-to-year variability in diameter at breast height (dbh) of trees. Briefly, the dbh of trees within a 20-m diameter plot were measured; all trees above 10 cm in diameter were sampled within 13 m and trees larger than 20 cm dbh were sampled in the remainder of the plot. In Valles Caldera, rather than subsampling within a 20-m plot, all trees were sampled from two central locations until fifty samples were collected from each location following Babst et al., (2014). At the Niwot site, a point-center-quarter method (Stearns, 1949; Cottam et al., 1953) was used to estimate stand density and to select individuals for sampling. Species, dbh and canopy position were recorded for each tree within the plots. Increment cores were dried, mounted, and sanded using standard dendrochronological procedures (Stokes and Smiley, 1968). Increments were first visually crossdated (Douglass, 1941) and then measured under a binocular microscope and statistically crossdated using COFECHA software (Holmes, 1983; Grissino-Mayer, 2001). Ring widths were scaled to dbh and allometric equations (Jenkins et al., 2004; Chojnacky et al., 2014) were applied to estimate biomass through time. When available site/region specific allometric equations were applied, and generalized species level allometric equations were used where these were not available. Trees that were sampled but lacked sufficient tree-ring data were gap-filled with a generalized additive mixed model to account for their biomass on the landscape (Alexander et al., in review). At Harvard and Howland, tree-ring reconstructed biomass was compared to biomass estimated from permanent plots established in 1969 and 1989 respectively; tree-ring biomass increment estimates fell within the 95% confidence intervals of biomass estimated from the permanent plots (Dye et al., 2016).





Biometric estimates of aboveground biomass were also available for some sites and years from the AmeriFlux network (Table 1). The $C_{stem}$/$C_{leaf}$ ratio, which was derived from AmeriFlux data with $C_{stem}$ and $C_{leaf}$ estimates for the same year, was only available for a subset of sites and years (Table 1).

In-situ measured LAI was available from AmeriFlux data for some sites (Table 1), and we used the annual maximum LAI for all the available measurements in each year. We used leaf C-LAI ratio from the AmeriFlux sites with simultaneous measurements of LAI and leaf C during the same year (Table 1).

### 2.3 C allocation scheme in CLM

The Community Land Model (CLM version 4.5) was used to simulate C fluxes, C pools and LAI at single points (PTCLM; Oleson et al., 2013). CLM is a component of the Community Earth System Model (CESM1.2) of the National Center for Atmospheric Research (Oleson et al., 2013).

CLM assumes that vegetated surfaces are comprised of different Plant Functional Types (PFTs). Our sites had two different PFTs: "needleleaf evergreen tree – temperate" for evergreen forests and "broadleaf deciduous tree – temperate" for deciduous forests.

CLM includes the following plant tissue types: leaf, stem (live and dead stem), coarse root (live and dead coarse root), and fine root. The model calculates carbon allocated to new growth based on three allometric parameters that relate allocation between tissue types (Oleson et al., 2013): *a1* (ratio of new fine root: new leaf carbon allocation); *a2* (ratio of new coarse root: new stem carbon allocation); and *a3* (ratio of new stem: new leaf carbon allocation). CLM has a dynamic allocation scheme (named "D-CLM"), which is described in Oleson et al. (2013), that includes one dynamic allometric parameter (as function of annual NPP) and two constant allometric parameters. In D-CLM (see Table 2), for the PFTs in our sites *a1* and *a2* are constant (*a1*=1, *a2*=0.3), whereas *a3* is a dynamic parameter defined by the following equation:

$$a3 = \frac{2.7}{1 + e^{-0.004*(NPPann-300)}} - 0.4$$

(1)

where *NPPann* is the annual sum of NPP of the previous year. The above equation for *a3* increases stem allocation relative to leaf when annual NPP increases. For instance, when annual NPP is 0 gCm$^{-2}$year$^{-1}$, *a3* is 0.20 (e.g. 0.2 units of C allocated to stem for 1 unit of C allocated to leaf), whereas when NPP is close to 1000 gCm$^{-2}$year$^{-1}$ or greater, *a3* is constrained to not exceed 2.2 (e.g. 2.2 units of C allocated to stem for 1 unit of C allocated to leaf). Therefore, when annual NPP is relatively close to 1000 gCm$^{-2}$year$^{-1}$ or greater the C allocation scheme becomes fixed with the following values for the parameters: *a1*=1, *a2*=0.3, and *a3*=2.2. For a broad range of annual NPP values, we calculated the allometric parameters *a1*, *a2* and *a3* and then converted the allometric parameters to allocation coefficients for each plant tissue using the C allometry in the model





(Oleson et al., 2013). We illustrate in one figure the effect of annual NPP on C allocation to each plant tissue in D-CLM (Fig. S1).

## 2.4 Alternative C allocation schemes

In addition to the dynamic C allocation scheme in CLM (Oleson et al., 2013), we implemented an alternative dynamic (Litton et al., 2007), and two fixed (Luyssaert et al., 2007) C allocation schemes.

The alternative dynamic C allocation scheme (named "D-Litton") was based on carbon partitioning data along an annual GPP gradient from Litton et al. (2007), and it considered two dynamic allometric parameters. We adapted the original equations reported in Litton et al. (2007), converted the GPP gradient to a NPP gradient with the general assumption that NPP=0.5×GPP

(Waring et al., 1998; Gifford, 2003), and used the modified equations to calculate the allometric parameters used in CLM. The partitioning between coarse root and fine root was not provided, and we used the default value for parameter *a1* (*a1*=1). The other allometric parameters (*a2* and *a3*) were dynamic, and the equations used for them are shown in Table 2.

The two alternative fixed C allocation schemes were based on observed values reported by Luyssaert et al. (2007), which were converted accordingly to the allometric parameters used in CLM. One of the C allocation schemes was representative of

temperate evergreen forests (named "F-Evergreen") and the other of temperate broadleaf deciduous forests (named "F-Deciduous"). Similarly to Litton et al. (2007), Luyssaert et al. (2007) only provided total root allocation without considering coarse and fine root, but the default value for parameter *a1* (*a1*=1) was not possible in some cases. We thus initially used a range of possible values for parameter *a1* (*a1*=1, *a1*=0.75 and *a1*=0.5) for model runs. When based on the values in Luyssaert et al. (2007) allocation to leaf was lower than total root allocation, we used the default value for parameter *a1* (*a1*=1 for F-

Evergreen); but when based on the values in Luyssaert et al. (2007) allocation to leaf was higher than total root allocation, the *a1* parameter had to be lower than 1. This was the case for the F-Deciduous C allocation scheme, and because *a1*=0.75 gave unrealistic aboveground:belowground ratios, we used *a1*=0.5. The allometric parameters used for the F-Evergreen and F-Deciduous C allocation scheme are shown in Table 2.

D-CLM and the alternative C allocation schemes have important differences in C allocation to each plant tissue (see Fig. S1).

Some of the main differences between D-CLM, and the alternative C allocation schemes, include increased allocation to leaf, and decreased allocation to stem, especially in D-Litton at sites with low mean annual NPP (see Fig. S1).

## 2.5 LAI in CLM

CLM uses a prognostic canopy model, with feedbacks between GPP and LAI acting through allocation to leaf C and Specific

Leaf Area (SLA) (Thornton and Zimmermann, 2007). The model assumes a linear relationship between SLA and the canopy depth (x):





$$SLA(x) = SLA_0 + mx \tag{2}$$

where $SLA_0$ (m$^2$ one-sided leaf area gC$^{-1}$) is a fixed value of SLA at the top of the canopy, $m$ is a linear slope coefficient, and $x$ is the canopy depth expressed as overlying leaf area index (m$^2$ overlying one-sided leaf area m$^{-2}$ ground area). LAI is

calculated for a given leaf C ($C_L$) using the following equation:

$$LAI = \frac{SLA_0[\exp(mC_L) - 1]}{m} \tag{3}$$

where $m$ and $SLA_0$ are different parameters for each PFT. In the case of temperate evergreen forests the default values for $m$

and $SLA_0$ in CLM are 0.00125 and 0.010; whereas for temperate broadleaf deciduous forests $m= 0.004$ and $SLA_0=0.030$ (Oleson et al., 2013).

We compared leaf C-LAI data from available sites with the leaf C-LAI relationship in the model. For deciduous sites, we optimized the model parameters based on observed leaf C-LAI. To avoid using unrealistic values for the parameters $m$ and $SLA_0$, we took a range of possible values for both parameters from Thornton and Zimmermann (2007), and used an optimization

approach that combined the range of parameter values and Eq. (3) to find the best combination of values for the two parameters given the leaf C-LAI observations at our sites. After optimizing the parameters $m$ and $SLA_0$, we used $m=0.0010$ and $SLA_0=0.024$ for deciduous forests. For evergreen sites, we could not optimize the parameters $m$ and $SLA_0$ due to the limited number of leaf C-LAI observations available.

**2.6 Turnover rate and aboveground biomass increment**

CLM, like many models, is based on differential equations for the calculation of changing biomass with time, which can be expressed as:

$$dB_i/dt = a_i\, NPP - u_i B_i \tag{4}$$

where $i$ is a given plant pool; $B_i$ is the biomass of that pool; $dB_i/dt$ is the biomass increment with time for each plant pool; $a_i$ is

the allocation coefficient to that plant pool (allocation coefficients for all pools combined sum to 1); and $u_i$ is the turnover rate for each component. We considered leaf, stem, coarse root and fine root as plant pools. To optimize the stem turnover rate we used a model emulator with the above equation to modify the default stem turnover rate (2%) to within a range of 0 to 2% (van Mantgem et al., 2009; Brown and Schroeder, 1999); for the rest of plant pools we used the default turnover rate in the model. In the model emulator, the annual NPP input was derived from the model for a given site using the default stem turnover (2%).



We compared the differences in aboveground biomass (leaf and stem) increment over time based on different turnover rates with the aboveground biomass increments estimated from tree-ring data for our sites between 1980 and 2011.

### 2.7 C allocation scheme effect on initial aboveground biomass and $C_{stem}/C_{leaf}$ ratio

5     The C allocation scheme used had a strong influence on initial aboveground biomass and the $C_{stem}/C_{leaf}$ ratio, which can be explained with Eq. (4). When the model is in equilibrium conditions, then $dB_i/dt=0$ in Eq. (4), and denoting $B_{stem}$ with $C_{stem}$, and $B_{leaf}$ with $C_{leaf}$:

$$a_{stem}NPP = NPP_{stem} = u_{stem}C_{stem} \tag{5}$$

$$a_{leaf}NPP = NPP_{leaf} = u_{leaf}C_{leaf} \tag{6}$$

After dividing Eq. (5) by Eq. (6):

$$a_{stem}/a_{leaf} = NPP_{stem}/NPP_{leaf} = \left(C_{stem}/C_{leaf}\right)\times\left(u_{stem}/u_{leaf}\right) \tag{7}$$

$$C_{stem}/C_{leaf} = \frac{\left(NPP_{stem}/NPP_{leaf}\right)}{\left(u_{stem}/u_{leaf}\right)} \qquad \text{or} \qquad C_{stem}/C_{leaf} = \frac{\left(a_{stem}/a_{leaf}\right)}{\left(u_{stem}/u_{leaf}\right)} \tag{8}$$

In D-CLM $NPP_{stem}/NPP_{leaf} \approx 2$ and $a_{stem}/a_{leaf} \approx 2$ for evergreen sites in favorable conditions (e.g. mean annual NPP $\approx 1000$ $gCm^{-2}year^{-1}$) and for deciduous sites; $u_{stem}/u_{leaf}=0.02$ for deciduous and $u_{stem}/u_{leaf}=0.06$ for evergreen forests. Therefore, in D-CLM $C_{stem}/C_{leaf} \approx 33$ for evergreen sites in favorable conditions; and $C_{stem}/C_{leaf} \approx 100$ for deciduous sites.

Because the alternative C allocation schemes have different $NPP_{stem}/NPP_{leaf}$ ratio than the one in D-CLM, they showed different $C_{stem}/C_{leaf}$ ratio, despite having the same $u_{stem}/u_{leaf}$ . We compared the $C_{stem}/C_{leaf}$ ratio from the four C allocation schemes with

available observations for the sites (Table 1).

In reference to the initial aboveground biomass (leaf+stem), we can use Eq. (4), and assuming equilibrium conditions, $dB_i/dt=0$, then:

$$a_{leaf}NPP + a_{stem}NPP = u_{leaf}C_{leaf} + u_{stem}C_{stem} \tag{9}$$





$$ANPP = u_{leaf}C_{leaf} + u_{stem}C_{stem} = C_{stem}(u_{stem} + u_{leaf}C_{leaf}/C_{stem}) \tag{10}$$

$$C_{stem}^* = ANPP/(u_{stem} + u_{leaf}C_{leaf}/C_{stem}) = ANPP/u_{stem}(1 + (NPP_{leaf}/NPP_{stem})) \tag{11}$$

Similarly to Eq (10),

$$ANPP = u_{leaf}C_{leaf} + u_{stem}C_{stem} = C_{leaf}(u_{leaf} + u_{stem}C_{stem}/C_{leaf}) \tag{12}$$

$$C_{leaf}^* = ANPP/(u_{leaf} + u_{stem}C_{stem}/C_{leaf}) = ANPP/u_{leaf}(1 + (NPP_{stem}/NPP_{leaf})) \tag{13}$$

Hence,

$$C_{aboveground}^* = C_{leaf}^* + C_{stem}^* = ANPP/u_{leaf}(1 + (NPP_{stem}/NPP_{leaf})) + ANPP/u_{stem}(1 + (NPP_{leaf}/NPP_{stem}))$$
$$\tag{14}$$

where $C_{stem}^*$, $C_{leaf}^*$, and $C_{aboveground}^*$ refer to stem C, leaf C and aboveground C in equilibrium conditions, respectively. Therefore, the aboveground biomass in equilibrium conditions will depend on aboveground NPP ($ANPP$), the $NPP_{stem}/NPP_{leaf}$ ratio (or $a_{stem}/a_{leaf}$ ratio) and the turnover rates for leaf and stem ($u_{leaf}$ and $u_{stem}$). We compared the effect of different C allocation schemes in initial aboveground biomass in equilibrium and we also compared them with tree-ring estimates of aboveground biomass data for 1980.

### 2.8 Model experiments

We used the CLM model – a well-established and commonly used LSM, as a platform to implement the alternative C allocation schemes, described above, and compared the resultant model simulations of C fluxes, C pools, LAI, and the $C_{stem}/C_{leaf}$ ratio with available observations. Four experiments were designed to better understand the impact of the different C allocation schemes. All modelling experiments were run for nine sites, including four evergreen and five deciduous forests (see Table 1). For evergreen sites, we used the default leaf C-LAI relationship in CLM, whereas for deciduous forests we used the optimized leaf C-LAI relationship (Sect. 2.5).

For experiment 1 we used the original dynamic C allocation scheme in CLM (D-CLM; see Sect. 2.3). For experiment 2, we used the alternative dynamic C allocation scheme based on Litton et al. (2007) (D-Litton, see Sect. 2.4). For experiments 3 and 4, we used a fixed C allocation scheme representative of evergreen (F-Evergreen) and deciduous (F-Deciduous) forests, respectively (Luyssaert et al. 2007 – see Sect. 2.4).





The standard climate forcing provided with the model is the 1901-2013 CRUNCEP dataset. The CRUNCEP dataset has been used to force CLM for studies of vegetation growth, evapotranspiration, and gross primary production (Mao et al., 2012; Mao et al., 2013; Shi et al., 2013; Chen et al., 2016), and for the TRENDY (trends in net land-atmosphere carbon exchange over the period 1980-2010) project (Piao et al., 2012).

In all the experiments, we spun-up the model for each site and C allocation scheme using 1901-1920 CRUNCEP climate and assuming pre-industrial atmospheric $CO_2$ concentration in order to bring all above- and belowground C pools to equilibrium. We used the initial conditions resulting from the spin-up to perform a 1901-2013 transient run (e.g. 1901-2013 CRUNCEP transient climate, transient atmospheric $CO_2$ concentration). Observations were compared with model outputs for the period

between 1980 and 2013.

## 3 Results

### 3.1 Carbon fluxes, and pools in D-CLM

When compared to observations from the AmeriFlux sites, D-CLM usually overestimated GPP (Fig. 1a), and ecosystem

respiration (Fig. 1c), and underestimated net ecosystem exchange (NEE; Fig. 1b).

Initial aboveground biomass in 1980 showed contrasting patterns in D-CLM for evergreen and deciduous forests. At evergreen sites, aboveground biomass in 1980 was underestimated at sites with mean annual NPP<500 $gCm^{-2}year^{-1}$ (NR1 and Vcm) and overestimated at the site with mean annual NPP>500 $gCm^{-2}y^{-1}$ (Ho1; Fig. 2a). Aboveground biomass in 1980 was largely overestimated at all deciduous sites (between 10527 and 12897 $gCm^{-2}$) (Fig. 2a). The accumulated aboveground biomass

between 1980 and 2011 was largely underestimated in the model (difference between observations and model ranged between 1222 and 7557 $gCm^{-2}$, depending on the site) (Fig. 2b).

### 3.2 LAI and $C_{stem}/C_{leaf}$ in D-CLM

D-CLM overestimated LAI relative to in-situ LAI measurements (Fig. 3a). We compared the leaf C-LAI relationship with the

observed leaf C-LAI and found important differences, especially for deciduous sites (Fig. 3b). We optimized the parameters $m$ and $SLA_0$ based on available observations for two deciduous sites (Fig. 3b). The modified LAI was closer to the LAI values measured in-situ for all five deciduous sites (Fig. 3c).

The $C_{stem}/C_{leaf}$ ratio in the model was dramatically different from the observations (Fig. 4). The model overestimated the $C_{stem}/C_{leaf}$ ratio in one of the two years with available data for two evergreen sites, and all the 19 years with available data for

two deciduous sites (Fig. 4; Table 1).



### 3.3 Carbon fluxes, pools and LAI in the alternative C allocation schemes

The accumulated annual C fluxes (GPP, ecosystem respiration, and NEE) from 1980 to 2011 gave comparable results for the four C allocation schemes (Suppl. Fig. 2). However, the C allocation schemes resulted in differences larger than 5000 $gCm^{-2}$ in long-term aboveground biomass accumulation for all the sites (Fig. 5a and 5b). All C allocation schemes overestimated

aboveground biomass in 1980 in all the sites, except in evergreen sites with mean annual NPP<500 $gCm^{-2}year^{-1}$ (NR1 and Vcm), where only the F-Deciduous allocation overestimated aboveground biomass (Fig. 5a). The D-Litton allocation scheme underestimated aboveground biomass in 1980 at all evergreen sites and, despite overestimating it at all deciduous sites, this scheme gave the closest values to the observations (Fig. 5a). Similar results were found for mean aboveground biomass between 2002 and 2011 (Fig. 5b). Despite the differences in the total aboveground biomass, aboveground biomass annual

increment in all the C allocation schemes was lower than that estimated from tree-ring data and accumulated aboveground biomass between 1980 and 2011 was therefore strongly underestimated assuming a mortality rate of 2% $year^{-1}$ (Fig. 5c).

The C allocation schemes showed differences of up to 10% in allocation to leaf, which produced large differences in LAI values (from ~20 to ~4.5) between allocation schemes (Fig. 6). In particular the F-Deciduous allocation gave high and unrealistic LAI values at evergreen sites (LAI ~ 20; Fig. 6), where the leaf C-LAI relationship was not optimized. At deciduous

sites, using the optimized leaf C-LAI relationship, the highest LAI values were ~10 (Fig. 6). The F-Deciduous allocation had an allocation to leaf that was ~10% greater than the one in D-CLM; however, the F-Deciduous allocation scheme with optimized LAI gave very similar LAI values to the D-CLM without optimizing the leaf C-LAI relationship (Fig. 6).

### 3.4 Turnover rate and its effect on accumulated aboveground biomass through time

When using stem turnover rates lower than 2% $year^{-1}$ (the default value used in CLM), the modeled accumulated aboveground

biomass between 1980 and 2011 was closer to the observed values for all the C allocation schemes (Fig. 7a, 7b, 7c, 7d). Overall, the turnover effect (difference in accumulated aboveground biomass between minimum and maximum stem turnover rate) was relatively low in the evergreen sites with annual NPP<500 g $Cm^{-2}year^{-1}$ (between 1999 and 3928 $gCm^{-2}$ in D-CLM), but it was relatively high in evergreen sites with annual NPP>500 g $Cm^{-2}year^{-1}$ and in the deciduous sites (between 10779 and 14342 $gCm^{-2}$ in D-CLM) (Fig. 7a, 7b, 7c, 7d). Relative to D-CLM, the D-Litton scheme considerably reduced the turnover

effect for evergreen sites with annual NPP>500 g $Cm^{-2}year^{-1}$ and deciduous sites (between 6395 and 9543 $gCm^{-2}$), whereas the F-Deciduous scheme increased the turnover effect for evergreen sites with annual NPP<500 g $Cm^{-2}year^{-1}$ (between 5115 and 7130 $gCm^{-2}$).

### 3.5 C allocation scheme and its effects on $C_{stem}/C_{leaf}$ ratio and initial aboveground biomass

The $NPP_{stem}/NPP_{leaf}$ ratio was overestimated in D-CLM, and it caused overestimations in the $C_{stem}/C_{leaf}$ ratio, which ranged between 33 and 56 for deciduous sites (Fig. 4, Fig. 8a). For the range of annual NPP values at our sites (NPP<1500 $gCm^{-2}year^{-}$



[1]), the $NPP_{stem}/NPP_{leaf}$ ratio was the lowest in the D-Litton scheme (Fig. 8b), which therefore resulted in the lowest $C_{stem}/C_{leaf}$ ratios amongst the four C allocation schemes (Fig. 8a). The $C_{stem}/C_{leaf}$ ratios from the D-Litton scheme were also the closest to the observed values at all the sites with mean annual NPP> 500 g $Cm^{-2}year^{-1}$ (Fig. 8a).

Initial aboveground biomass showed different patterns between evergreen and deciduous sites (Fig. 9a, 9b). Whereas for
evergreen sites with annual NPP<500 g $Cm^{-2}year^{-1}$, there was some overlap between modeled and observed initial aboveground biomass, for deciduous sites modeled initial aboveground biomass was strongly overestimated (between 10527 and 12897 $gCm^{-2}$) in D-CLM (Fig. 5a, Fig. 9b). The D-Litton scheme reduced the initial aboveground biomass relative to D-CLM, but still with a positive bias (between 5040 and 6859 $gCm^{-2}$) (Fig 5a, Fig. 9b).

## 4 Discussion

From the four C allocation schemes used, two were based on fixed coefficients (Luyssaert et al., 2007), whereas the other two were dynamic based on optimisation of resources (Oleson et al., 2013; Litton et al., 2007). Of these schemes, the dynamic scheme based on D-Litton performed better than the other three. Though this scheme is imperfect, we note that on average it produces lower, and more credible initial aboveground biomass estimates for these forests (Fig. 5a) and matches the biometric
estimates of C partitioning between leaf and stem (Fig. 8a). The evergreen and deciduous forests appear to allocate carbon differently and for situations where a fixed scheme is preferred our results favour the adoption of separate schemes for evergreen and deciduous forests. Below we discuss these findings in detail and make some recommendations for future development of allocation schemes.

## 4.1 C allocation scheme: implications for C flux, C pools and LAI

The C allocation scheme does not strongly influence annual GPP, ecosystem respiration, and NEE over 34 years of accumulated effect (Fig. S2); the general over-estimate of GPP and ecosystem respiration in Fig. 1 was common to all allocation schemes. GPP was also overestimated in previous versions of CLM (Bonan et al., 2011; Lawrence et al., 2011). Despite revisions of the model structure in previous versions of CLM, and that the GPP bias was found to be most pronounced
in the tropics (Lawrence et al., 2011), our results show that the GPP is still overestimated in temperate forests with the current version of CLM (CLM4.5).

When comparing estimated and modeled aboveground biomass values for the different sites, we found contrasting patterns for evergreen and deciduous forests. D-CLM underestimated the modeled aboveground biomass for evergreen sites with mean annual NPP<500 g $Cm^{-2}year^{-1}$, but overestimated it for deciduous sites. These results are in line with previous findings in
evergreen Oregon forests where CLM also underestimated aboveground biomass at most sites (Hudiburg et al., 2013). In a comparison between observations and CMIP5 Earth System Models for tropical forests, the high CLM-based biomass values





were attributed to the high stem allocation relative to observations (Negron-Juarez et al., 2015). A similar pattern has been found in other models, such as IAP-DGVM1.0, which also had a high allocation to stem that resulted in an overestimated aboveground biomass (Song et al., 2016). Our results support this point: our temperate deciduous sites, which generally had a higher mean annual NPP and therefore a higher allocation to stem in D-CLM than our evergreen sites, showed a strong

overestimation of aboveground biomass. Our results show that an alternative scheme (D-Litton, based on Litton et al., 2007), which greatly reduced allocation to stem compared with D-CLM, provided more realistic estimates of aboveground biomass for deciduous sites (Fig. 5a and 5b). However, the D-CLM-based estimates of aboveground biomass were closer to the observed values than those from the D-Litton scheme for evergreen sites with mean annual NPP<500 g $Cm^{-2}year^{-1}$ (NR1 and Vcm). Our results suggest that it is necessary to improve the D-CLM scheme for temperate forests, and that the D-Litton

scheme can be modified adapting the equations used here to non-linear equations to increase allocation to stem for sites with mean annual NPP<500 g $Cm^{-2}year^{-1}$.

When compared to syntheses of temperate forests, LSMs tend to underestimate allocation to leaves and overestimate allocation to stem. We designed the D-Litton, F-Deciduous and F-Evergreen schemes to match recent syntheses (Table 3). However, C allocation to leaf in D-CLM is probably underestimated when mean annual NPP is relatively close to or greater than 1000 g

$Cm^{-2}year^{-1}$. In other LSMs carbon allocation to leaf shows broad ranges (~19-30%; Table 3; Ise et al., 2010; Xia et al., 2015). The D-CLM scheme is dynamic with C but functions as a fixed scheme at higher NPP values (Fig. S1) which means that at many sites allocation to leaf is 20% in this scheme, which is ~5-10% lower than available data suggests for deciduous sites (Table 3; Litton et al., 2007; Luyssaert et al., 2007; Wolf et al., 2011). Similarly, D-CLM stem C allocation has a value of ~46% when annual NPP is close to or greater than 1000 g $Cm^{-2}year^{-1}$, while forest data syntheses indicate that 20-35% are

more plausible for sites with similar mean annual NPP to our sites (Litton et al., 2007). Other LSMs have an even higher allocation to stem of 45-50% for temperate forests (Table 3; Ise et al., 2010, Xia et al., 2015).

There is reasonable agreement across LSMs on how much carbon is allocated to roots, however root biomass is difficult to measure accurately and data are rare. Allocation to root and stem are variable between sites, and conditions that favour high productivity increase partitioning to stem and decrease partitioning to root (Litton et al., 2007). D-CLM allocates 34-40% of

carbon belowground which is similar to other models (Table 3), though notably larger than IBIS (~20%; Xia et al., 2015). The partitioning between fine and coarse root is absent from most syntheses but empirical studies show a wide range in allocation of C belowground and are generally higher than LSMs (Table 3; Nadelhoffer and Raich, 1992, Gower et al., 2001, Newman et al., 2006, Luyssaert et al., 2007, Litton et al., 2007, Wolf et al., 2011; Gill and Finzi, 2016).

Our results support the recommendation by Thornton and Zimmerman (2007) that additional measurements are required to

establish the variability of SLA(x) within and between PFTs. Maximum LAI values reported for temperate evergreen and deciduous forests are 15 and 8.8, respectively (Asner et al., 2003). The standard leaf C-LAI relationship resulted in unrealistically high – sometimes >20 – estimates of maximum annual LAI values when implementing alternative C allocation schemes in CLM. When using the optimized parameters in conjunction with the alternative allocation schemes, LAI always remained below 10. Realistic C allocation schemes (e.g. Litton et al., 2007; Luyssaert et al., 2007) in CLM combined with the





default values for the parameters $SLA_0$ and $m$ can give unrealistic LAI values. Unrealistic simulations of LAI also had to be addressed prior to using aboveground biomass data to optimize allocation parameters in the ORCHIDEE LSM (Thum et al., 2017). Site specific estimates of SLA and LAI would be very useful for optimizing parameters within their observed range and allow mechanistic processes controlling allocation to leaves in the model to be assessed.

Although in reality, root function is extremely dynamic, the controls of root dynamics and function are highly simplified in LSMs (Warren et al., 2015). It has been suggested that the functional trade-off hypothesis (Tilman, 1988) does not occur directly as a trade-off between leaf and fine root, but instead from two separate trade-offs between leaf or fine roots and their supporting woody organs (Chen et al., 2013). If this is correct, LSMs should use an allocation scheme based on at least two (or probably three) dynamic allometric parameters, instead of the D-CLM which is based only on one dynamic allometric

parameter ($a3$). Here, we implemented an allocation scheme (D-Litton) that included two dynamic allometric parameters ($a2$ and $a3$) based on Litton et al., (2007), assuming that the ratio between allocation to leaf and fine root ($a1$) is constant. However, some studies suggest that this trade-off includes fine roots (Wolf et al., 2011; Malhi et al., 2011; Chen et al., 2013), probably due to the co-limitation of productivity by resources captured aboveground (e.g. light) and belowground (e.g. nutrients and water) (Dybzinski et al., 2011). Furthermore, this complexity is enhanced by the fact that the relative influences of the growth

drivers strongly vary with time and across spatial ecological gradients (Guillemot et al., 2015). In the version of CLM (CLM4.5) employed here, the roots control water uptake but are not related to nutrient uptake. Understanding the mechanisms responsible for these multiple trade-offs and integrating them in the C allocation schemes of models is critical for accurate predictions of changes in carbon sequestration, including $CO_2$ impacts on forest productivity and allocation (De Kauwe et al., 2014; Hickler et al., 2015; Sevanto and Dickman 2015), and for determining the extent of atmospheric $CO_2$ accumulation in

the coming decades (Atkin, 2016). Root functionality in LSMs could be enhanced by improving parameterization within models and introducing new components such as dynamic root distribution and root functional traits linked to resource extraction (Warren et al., 2015; Brzostek et al., 2014; Shi et al., 2016; Phillips et al., 2016; Brzostek et al., 2017; Iversen et al., 2017). More process based root dynamics in LSMs could enable functional trade-offs to be used as a method to constrain allocation to roots.

### 4.2 C allocation scheme: implications for steady state aboveground biomass

Initial conditions in LSMs are usually obtained by spin-up methods that perform long simulations until the model reaches a steady state, a point when C pool sizes remain constant over long periods of repeated climate forcing (Xia et al., 2012). The simulation critically depends on the initial values and flawed initial conditions may produce a model output that can be severely

biased or unrealistic (Yang et al., 1995; Cosgrove et al., 2003; Rodell et al., 2005; Li et al., 2009). There is an increasing awareness in Earth system modeling of the critical role of initial conditions in model behaviour that adds an extra layer of complexity in diagnosing the impact of an incorrect representation of physical processes on the transient simulation (Kay et al., 2015; Fisher et al., 2015). Our results reinforce that concern by showing that with the same climate forcing different C



allocation schemes within the same LSM can produce strongly differing initial conditions for aboveground biomass (Fig. 9). In Sect. 2.7, we provide an explanation for the variability in steady state aboveground biomass depending on the C allocation scheme used in CLM.

## 4.3 C allocation scheme: implications of the $NPP_{stem}/NPP_{leaf}$ ratio

The $NPP_{stem}/NPP_{leaf}$ ratio (*a3* parameter) used in CLM has two important implications. Firstly, for the residence time given for the plant pools in CLM, the $NPP_{stem}/NPP_{leaf}$ ratio in D-CLM is causing an overestimation of $C_{stem}/C_{leaf}$ ratio when compared to observations (see also Sect. 2.7). We show that it is possible to simulate more realistic $C_{stem}/C_{leaf}$ ratios in CLM by decreasing the D-CLM $NPP_{stem}/NPP_{leaf}$ ratio in the model from values >2 to values ~1 or ~1.25, similar to the values in the D-Litton scheme (see Fig. 8b). The second implication is that if CLM overestimates the $NPP_{stem}/NPP_{leaf}$ ratio, it will also overestimate aboveground biomass due to the long residence time of stem (Schulze et al., 2000; Xia et al., 2015; Song et al., 2016). We also found important overestimations of aboveground biomass for deciduous forests with D-CLM, and therefore suggest that the $NPP_{stem}/NPP_{leaf}$ ratio in the model is one of the primary factors contributing to these overestimations of biomass. Although several ecosystem models (e.g. Hyland, IBIS, Biome-BGC, VISIT) allocate most of the carbon to stem for deciduous forests (Xia et al., 2015), allocation to stem was considerably reduced after constraining the allocation parameters in the model with satellite data (Xia et al., 2015). Similarly, our results suggest that allocation to stem in D-CLM should decrease, whereas allocation to leaf and root should increase, in order to align simulated and observed biomass.

## 4.4 C allocation scheme and residence time for stem: implications for accumulated aboveground biomass

When comparing average annual aboveground biomass increment derived from the four C allocation schemes with aboveground biomass increments reconstructed from tree rings for the period between 1980 and 2011, we found that it was underestimated at all sites. The underestimation can be attributed to an inaccurate representation of production in the model, an inaccurate representation of turnover time of the plant pools in the model, or both (Friend et al., 2014; Koven et al., 2015). For deciduous sites, when comparing aboveground NPP in the D-CLM scheme with available aboveground NPP from some of our sites, including UMBS, Morgan Monroe, Harvard Forest, and Duke hardwoods (Megonigal et al., 1997; Curtis et al., 2002), the model consistently overestimated aboveground NPP relative to the observations. The D-Litton scheme, however, resulted in aboveground NPP estimations that were consistently closer to the observations (data not shown). These results suggest that, in temperate deciduous forests, the D-CLM scheme is overestimating allocation to stem, and underestimating allocation to roots, as previously found in other models like IBIS (Xia et al., 2015).

Given that the model overestimates aboveground NPP and underestimates aboveground biomass increments, this suggests that the stem turnover rate is overestimated in the model. Given the high uncertainty associated with turnover relative to production, it has been suggested that research priorities should move from production to turnover (Friend et al., 2014). It is possible that



CLM - at least for deciduous sites - overestimated aboveground NPP as well as stem turnover. Turnover of biomass in forests includes annual loss of leaf, root and woody litter as well as tree mortality. These turnovers influence C residence time, a key factor that determines C storage capacity, but it is not well constrained in models (Friend et al., 2014; Chen et al., 2015). Tree-ring widths are measured with high precision and can thus result in reliable estimates of biomass increment (Alexander et al., in review; Dye et al., 2016; Klesse et al., 2016; Babst et al., 2014), but turnover is difficult to estimate from these data because of how they are influenced by stand age and disturbance history. The Harvard Forest, for example, is at the end of the stem exclusions stage; some secondary regeneration has begun. And, there has been little to no canopy disturbance since the time of the 1969 census. Thus, most of the mortality is self-thinning or thinning from below and the canopy has been stable. The loss of most trees through self-thinning are relatively small loses in terms of biomass and competition. As such, the tree-ring biomass increment estimates at Harvard and Howland assume zero mortality between 1980 and 2012, resulting in no significant difference between tree-ring reconstructed biomass increment and the repeated measurements from permanent plots over the last 40 years (Dye et al., 2016). Currently, CLM assumes a stem mortality rate of 2% $yr^{-1}$ that is higher than published tree mortality rates for forests in the USA (van Mantgem et al., 2009; Brown and Schroeder, 1999; Runkle, 1998). When considering whole ecosystem C turnover over large geographic scales the 2% $yr^{-1}$ rate of stem turnover may be reasonable. If we assume that tree-ring increment is a good proxy for biomass increment over this time window (Dye et al., 2016; Klesse et al., 2016), and the model captures the observed biomass increment from tree-rings, then the model can be used to estimate reasonable turnover rates for stems. Over large geographic scales a 2% $yr^{-1}$ stem turnover rate may be reasonable. However, whole ecosystem C turnover will encompass processes other than mortality, including disturbances, land use and land cover change (Masek et al., 2008; Erb et al., 2016; Thurner et al., 2017) – such processes are partially incorporated in LSMs, but some impacts of these processes are also implicitly represented in stem turnover rates. We thus decreased stem mortality rate from 2% $yr^{-1}$ to published ranges of tree mortality (between 0 and 1.5% $yr^{-1}$), and the resulting ranges of aboveground biomass increment included the observed aboveground biomass increment, which was estimated from tree-ring data, for all the carbon allocation schemes (see Fig. 7a, 7b, 7c, and 7d) except in evergreen sites with mean annual NPP<500 g $Cm^{-2}year^{-1}$ with the D-Litton scheme. This suggests that D-Litton is underestimating aboveground NPP at these sites as pointed out in Sect. 4.2 (Fig. 7b). However, a different turnover rate was required for each site and C allocation scheme to match the observed aboveground biomass increment. Our analysis suggests that when using AmeriFlux sites to inform models, or other site level observations, taking note of site specific rates of stem turnover is prudent. Our results show the need for improvements of models in carbon turnover processes, a current limitation in state-of-the-art LSMs (Thurner et al., 2017). Furthermore, we should be clear what we are referring to when considering turnover rate in the models, and be careful not to use this parameter to account for missing processes or scaling issues (Thum et al., 2017).



### 4.5 Conclusions and perspectives

Our results highlight the importance of evaluating the C allocation scheme and the stem turnover in LSMs using biometric data in addition to flux data. The four C allocation schemes translated to important long-term differences in C accumulation in aboveground biomass, but gave similar results for short term C fluxes. There is no way to distinguish between the allocation

schemes using eddy flux data alone.

Developing allocation schemes for LSMs is challenging. The two dynamic allocation schemes reflect forest stand development to some extent i.e. as trees get bigger (and can grow more) they tend to invest more in stem and less in leaves. The two schemes also use low NPP as a proxy for resource limitation, but they disagree on how allocation changes as a function of NPP (Fig. S1). However, these schemes and many other LSMs do not have a way to consider cohorts of trees. This problem is highlighted

in the different performance of the D-CLM scheme at high and low NPP; sites that have low NPP perpetually allocate more resources to leaves and roots while sites with high NPP perpetually allocate less resources to leaves and roots (Fig. S1). This increases the NPPstem/NPPleaf ratio with increasing NPP (Fig. 8b), and it causes the overestimation of the Cstem/Cleaf ratio relative to observations (Fig. 8a) at most of the sites (except at low NPP sites; NPP<500 gC $m^{-2}year^{-1}$). Ecological theory suggest that dynamic allocation probably reflects whatever resource is most limiting. As coupled C-N and functional root

subroutines are developed for LSMs with better representation of vegetation dynamics (Fisher et al., 2015), we could imagine a dynamic allocation scheme based on whether above ground (light) or below ground (water and nutrients) are limiting.

Data on different carbon pools is sparse, but very useful in parameterizing the non-physiological components of LSMs. We found that site specific SLA was a pre-requisite to evaluating the different allocation schemes; large scale databases might be exploited to better estimate this relationship. Also, fixed allocation schemes are unable to capture dynamic changes in

allocation in response to varying water and nutrient availability at seasonal to interannual timescales (De Kauwe et al., 2014) but they have the advantage of simplicity. If fixed allocation schemes are used in land surface modelling, we suggest different schemes for evergreen and deciduous forests, and that databases like Litton et al. (2007) and Luyssaert et al. (2007) can be used to parameterize them.

Finally, we show that information on forest age and successional status is important to interpret the success or failure of

different model schemes at forest sites. Some aspects of LSMs are most consistent with ecological processes that may approximate steady state conditions at large scales, and so are inconsistent with forests which are not at steady state. Decreasing the stem turnover rate from 2% $yr^{-1}$ to plausible values consistent with their successional status yielded aboveground biomass accumulation rates more consistent with observations.  It is possible to coarsely estimate equilibrium turnover rates from mean stand age; this could be a promising technique to more firmly estimate carbon residence times in temperate forests.





**Code availability**

The code for CLM version 4.5 (CLM4.5) is available (registration required) at https://svn-ccsm-models.cgd.ucar.edu/cesm1/release_tags/cesm1_2_1. The allometric parameters used for the different C allocation schemes used in this study with CLM are available in Table 2. The optimized parameters, based on observations, for the leaf C-LAI

5    relationship for temperate deciduous forests in CLM are available in Sect. 2.5. The code for this paper is available upon request, contacting the corresponding author.

**Data availability**

The data for this paper is available upon request, contacting the corresponding author.

*Acknowledgements.*

This study was supported by the DOE Regional and Global Climate Modeling DE-SC0016011.

The US-NR1 and US-MMS AmeriFlux sites are currently supported by the US DOE, Office of Science, through the AmeriFlux Management Project (AMP) at Lawrence Berkeley National Laboratory under award numbers 7094866 and 7068666,

respectively.

AmeriFlux site US-MOz is supported by the U.S. Department of Energy, Office of Science, Office of Biological and Environmental Research Program, through Oak Ridge National Laboratory's Terrestrial Ecosystem Science (TES) Science Focus Area (SFA). ORNL is managed by UT-Battelle, LLC, for the U.S. DOE under contract DE-AC05-00OR22725.

FB acknowledges funding from the Swiss National Science Foundation (#P300P2_154543) and the EU-Horizon 2020 Project

"BACI" (#640176).

MRA was supported by the DOE Regional and Global Climate Modeling program DE-577SC0016011 and by the University of Arizona Water, Environment, and Energy Solutions (WEES) 578 and Sustainability of Semi-Arid Hydrology and Riparian Areas (SAHRA) programs.

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

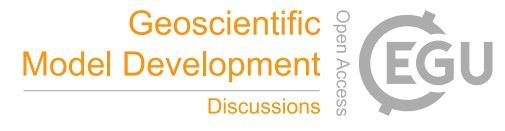

**Table 1.** Site general information and observations available. ID refers to site name used in the AmeriFlux network.

| Site (ID) | Longitude | Latitude | Reference | C fluxes data | Aboveground biomass (AmeriFlux) | Aboveground biomass (tree-ring) | LAI in-situ data | LAI data | Leaf C-LAI data | Stem C/Leaf C data |
|---|---|---|---|---|---|---|---|---|---|---|
| **Evergreen** | | | | | | | | | | |
| Niwot Ridge (NR1) | -105.5464 | 40.0329 | Blanken, 2016 | 1999-2013 | 2003 | 1980-2012 | n.a. | n.a. | n.a. | n.a. |
| Valles Caldera Mixed Conifer (Vcm) | -106.5321 | 35.8884 | Litvak, 2016 | 2007-2013 | 2007 | 1980-2011 | n.a. | n.a. | n.a. | 2007 |
| Howland Forest (Ho1) | -68.7402 | 45.2041 | Hollinger, 2016 | 1996-2004 2006-2013 | 2003 | 1980-2012 | 2006 | n.a. | n.a. | 2003 |
| Duke Forest Loblolly Pine (Dk3) | -79.0942 | 35.9782 | Stoy et al., 2016 | 1998-2005 | 2001-2005 | n.a. | 2002-2005 | 2002-2005 | 2002-2005 | n.a. |
| **Deciduous** | | | | | | | | | | |
| University of Michigan Biological Station (UMB) | -84.7138 | 45.5598 | Gough et al., 2009; Gough et al., 2013; Gough et al., 2016 | 2005-2013 | 1998-2011 | 1980-2013 | 1997-2013 | 2009 | 1998-2009 | 1998-2009 |
| Harvard Forest (Ha1) | -72.1715 | 42.5378 | Munger, 2016 | 1992-2013 | 2006-2008 | 1980-2012 | 1998,1999, 2005-2008, 2010 | n.a. | n.a. | n.a. |




| Site | Longitude | Latitude | Reference | | | | | | |
|---|---|---|---|---|---|---|---|---|---|
| Missouri Ozark (MOz) | -92.2000 | 38.7441 | Wood and Gu, 2016 | n.a. | n.a. | 1980-2013 | 2006-2012 | n.a. | n.a. |
| Morgan Monroe State Forest (MMS) | -86.4131 | 39.3232 | Novick and Phillips, 2016 | 1999-2013 | 1999-2005 | 1980-2013 | 1999-2006, 2009 | 1999-2005 | 1999-2005 |
| Duke Forest Hardwoods (Dk2) | -79.1004 | 35.9736 | Oishi et al., 2016 | 2001-2005 | 2002 | 1980-2013 | 2006 | n.a. | n.a. |



**Table 2.** Allometric parameter values for evergreen and deciduous temperate forests in the C allocation scheme in CLM described in Oleson et al. (2013) (D-CLM); the alternative dynamic C allocation scheme (D-Litton) based on Litton et al. (2007); and the 2 fixed C allocation schemes (F-Evergreen, and F-Deciduous) based on Luyssaert et al. (2007). Allometric parameters represented with numbers indicate constant parameters, whereas equations indicate dynamic parameters. In the equations, *NPPann* is the annual sum of Net Primary Productivity (NPP) of the previous year.

| Allometric parameter | Definition (parameter name) | C allocation scheme | | | |
|---|---|---|---|---|---|
| | | D-CLM | D-Litton | F-Evergreen | F-Deciduous |
| a1 | Ratio of new fine root: new leaf carbon allocation (froot_leaf) | 1 | 1 | 1 | 0.5 |
| a2 | Ratio of new coarse root: new stem carbon allocation (croot_stem) | 0.3 | $\dfrac{0.25 - 8e^{-05} \times NPPann}{0.17 + 0.0001158 \times NPPann}$ | 0.27 | 0.27 |
| a3 | Ratio of new stem: new leaf carbon | $\dfrac{2.7}{1 + e^{-0.004 \times (NPPann-300)}} - 0.4$ | $\dfrac{0.17 + 0.0001158 \times NPPann}{0.26}$ | 1.76 | 1.4 |



allocation
(stem_leaf)



**Table 3.** Percentage of NPP allocated to the each plant pool (leaf, stem, and belowground) according to observations, the four C allocation schemes used (D-CLM, D-Litton, F_Evergreen, and F-Deciduous), and C allocation schemes of other models.

| % Allocation | Observation (Reference) | C allocation scheme | | | | Other models (Model; Reference) |
| --- | --- | --- | --- | --- | --- | --- |
| | | D-CLM | D-Litton | F-Evergreen | F-Deciduous | |
| % Leaf | ~25-30% (Luyssaert et al., 2007) / ~26% (Litton et al., 2007) / ~25% (Wolf et al., 2011) | ~30% in low NPP sites / ~20% in high NPP sites | ~26% | ~25% | ~30% | 19.8% (VISIT; Ise et al. 2010) / 19% for evergreen and 20% for deciduous (BIOME-BGC; Ise et al., 2010) / 30% (IBIS; Xia et al., 2015) |
| % Stem | ~41-43% (Luyssaert et al., 2007) / ~20-35% in sites with similar NPP to the sites in this study (Litton et al., 2007) / ~20-35% (Wolf et al., 2011) / ~30-38%; assuming NPP=0.5×GPP (Chen et al., 2013) | ~25% in low NPP sites / ~46% in high NPP sites | ~20-35% | ~41% | ~43% | 50% (VISIT; Ise et al. 2010) / 42% for evergreen and 45% for deciduous (BIOME-BGC; Ise et al., 2010) / 50% for temperate broadleaf forests (IBIS; Xia et al., 2015) |
| % Belowground (fine root+coarse root) | ~34-37% (Luyssaert et al., 2007) / ~39-54% in sites with similar NPP to the sites in this study (Litton et al., 2007) | ~40% in low NPP sites | ~39-54% | ~34% | ~37% | 30.2% (VISIT; Ise et al. 2010) / 39% for evergreen and 34% for deciduous (BIOME-BGC; Ise et al., 2010) / 20% (IBIS; Xia et al., 2015) |



~50% for temperate forests (Newman et al., 2006)

~45-50% as mean values for temperate forests (Gill and Finzi, 2016)

~34% in high NPP sites



# FIGURES

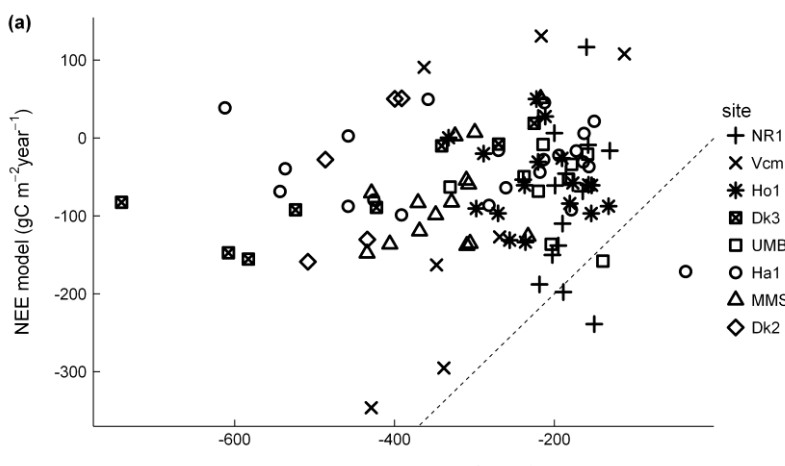

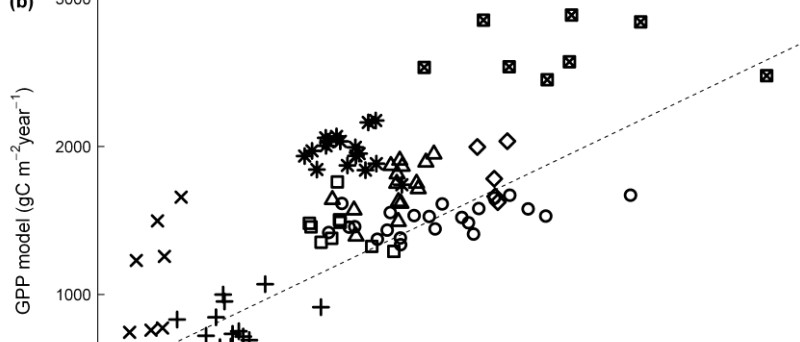

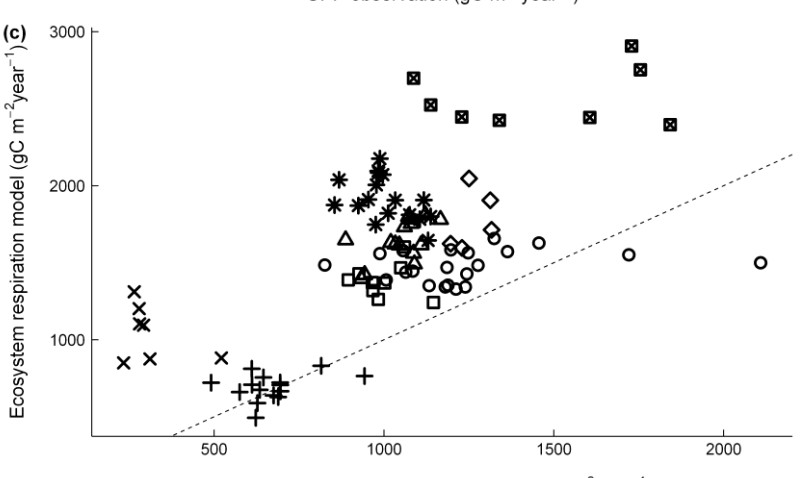





**Figure 1** Comparisons between (a) NEE, (b) GPP and (c) ecosystem respiration in observations and model (D-CLM). All fluxes were aggregated to annual. Dashed line is 1:1 relationship between observations and model. Observations are from the AmeriFlux L2 data product.

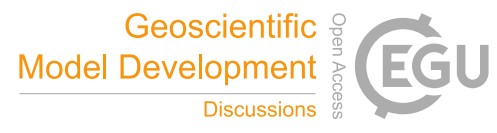

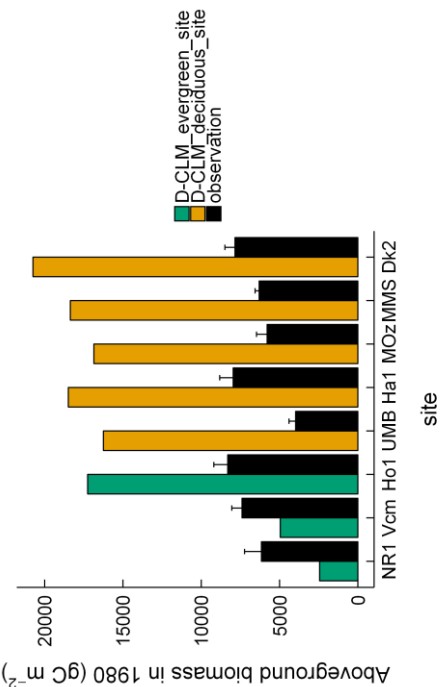

**Figure 2** Comparisons between (a) observed and modeled (D-CLM) aboveground biomass in 1980; (b) observed and modeled (D-CLM) accumulated aboveground biomass between 1980-2011. Dashed line is 1:1 relationship between observations and model. Observations (estimates of aboveground biomass from tree-ring data) for the Ho1 and Ha1 sites are from Dye et al. (2016), whereas for the rest of sites observations were obtained following the methodology described in Alexander et al. (under review).



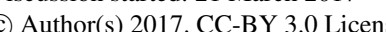




**Figure 3** Comparisons between (a) LAI measured in-situ and LAI in the model; (b) Relationship between Leaf C and LAI in:

5  CLM for deciduous forests, observations for deciduous forests, optimized Leaf C-LAI relationship for deciduous forests, CLM for evergreen forests, and observations for evergreen forests; (c) Comparisons between LAI measured in-situ and LAI in the standard and modified version of the model with optimized parameters for LAI. In 3a and 3c, dashed line is 1:1 relationship between observations and model. Observations (LAI measured in-situ, and Leaf C) are from the AmeriFlux database.





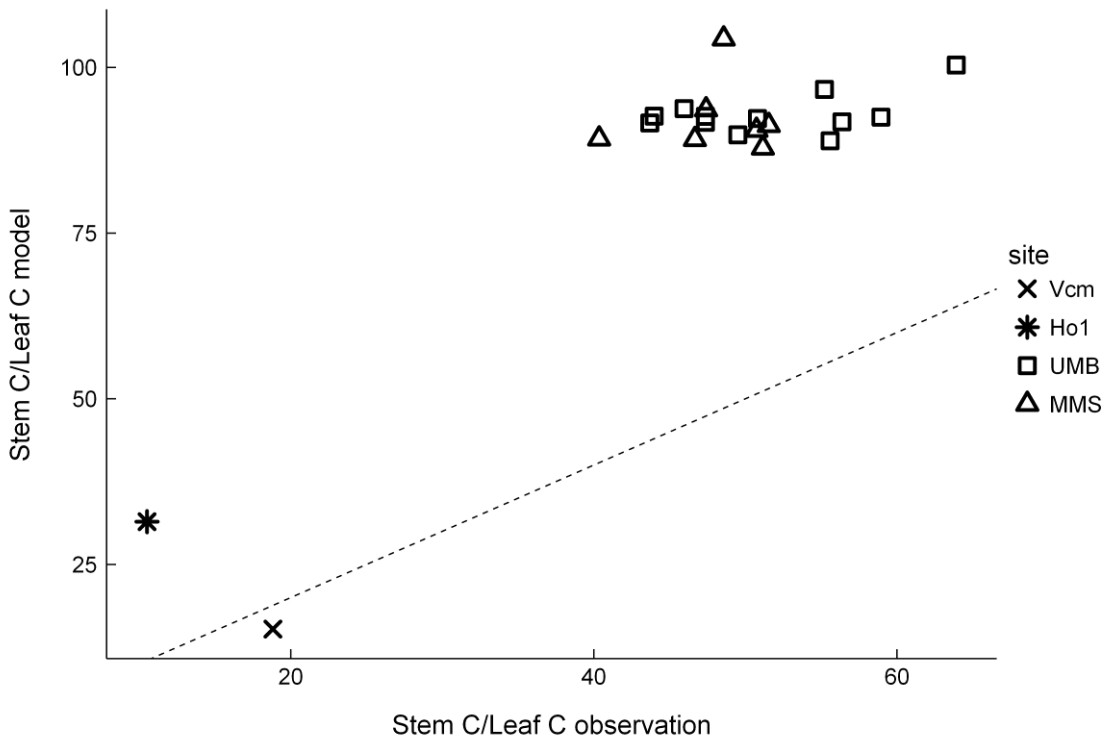

**Figure 4** Comparisons between Cstem/Cleaf ratio for the D-CLM scheme and AmeriFlux observations. Dashed line is 1:1 relationship between observations and model. Observations (Cstem and Cleaf) are from the AmeriFlux database.





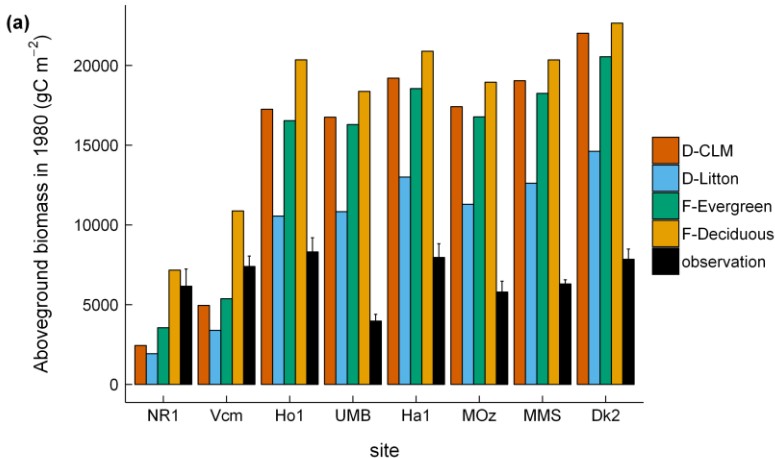

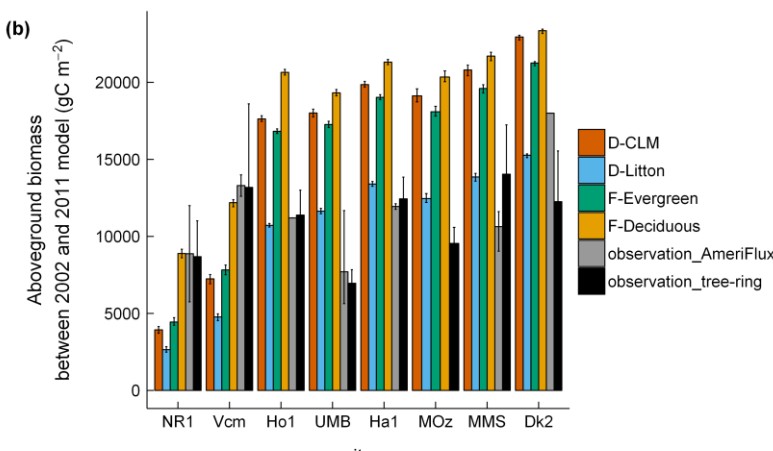

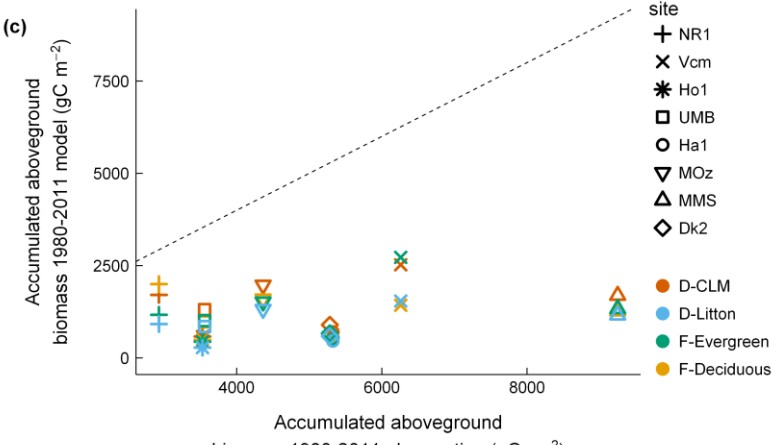





**Figure 5** (a) Comparisons between observed and modeled aboveground biomass in 1980 for the four C allocation schemes;
(b) Comparisons between mean observed and modeled aboveground biomass between 2002 and 2011 for the four C allocation
schemes; (c) Comparisons between observed and modeled accumulated aboveground biomass 1980-2011 for the four C
allocation schemes. Turnover rate for stem in CLM is 2%. Dashed line is 1:1 relationship between observations and model.
Observations ("observation" in 5a, "observation_tree_ring" in 5b, and "accumulated aboveground biomass 1980-2011
observation" in 5c) are aboveground biomass estimates from tree-ring data, which are from Dye et al. (2016) for the Ha1 and
Ho1 sites, and following the methodology in Alexander et al. (under review) for the rest of sites. Observations
("Observation_AmeriFlux" in 5b) are aboveground biomass data from the AmeriFlux database, available only for a subset of
sites and years (see Table 1).

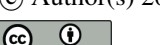


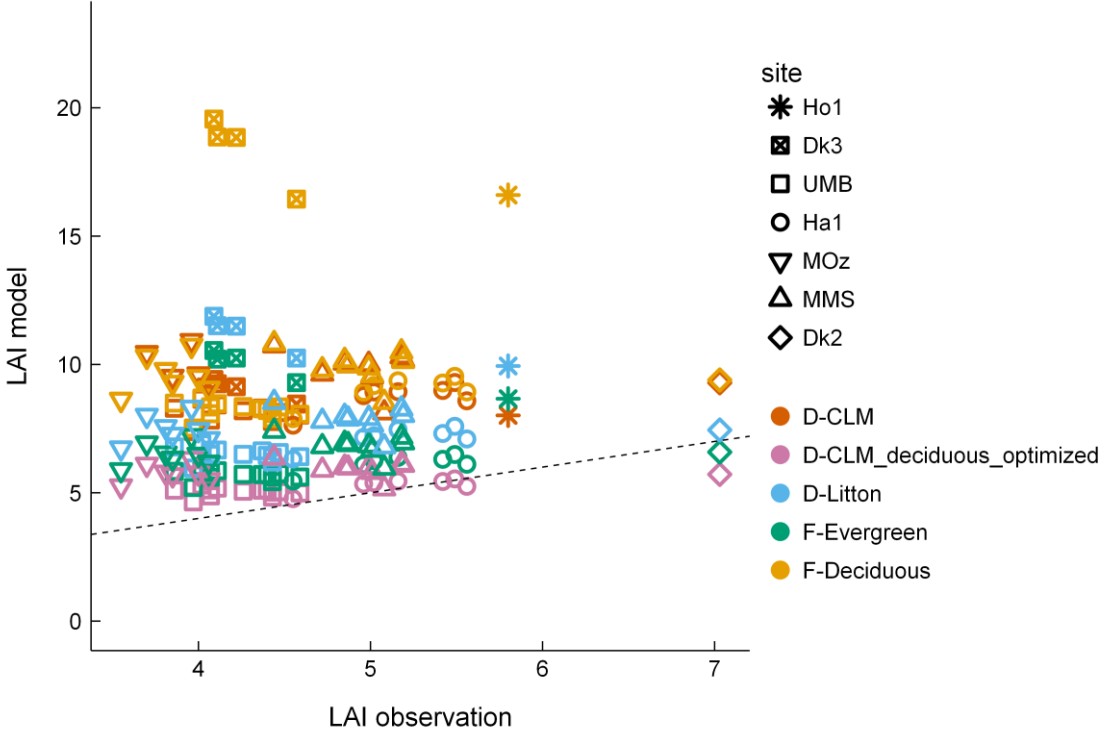

**Figure 6** Comparisons between LAI measured in-situ and LAI in the model for the different C allocation schemes (D-CLM_deciduous_optimized refers to the one with the optimized leaf C-LAI relationship for deciduous forests in D-CLM). Dashed line is 1:1 relationship between observations and model. Observations (LAI measured in-situ) are from the AmeriFlux database.

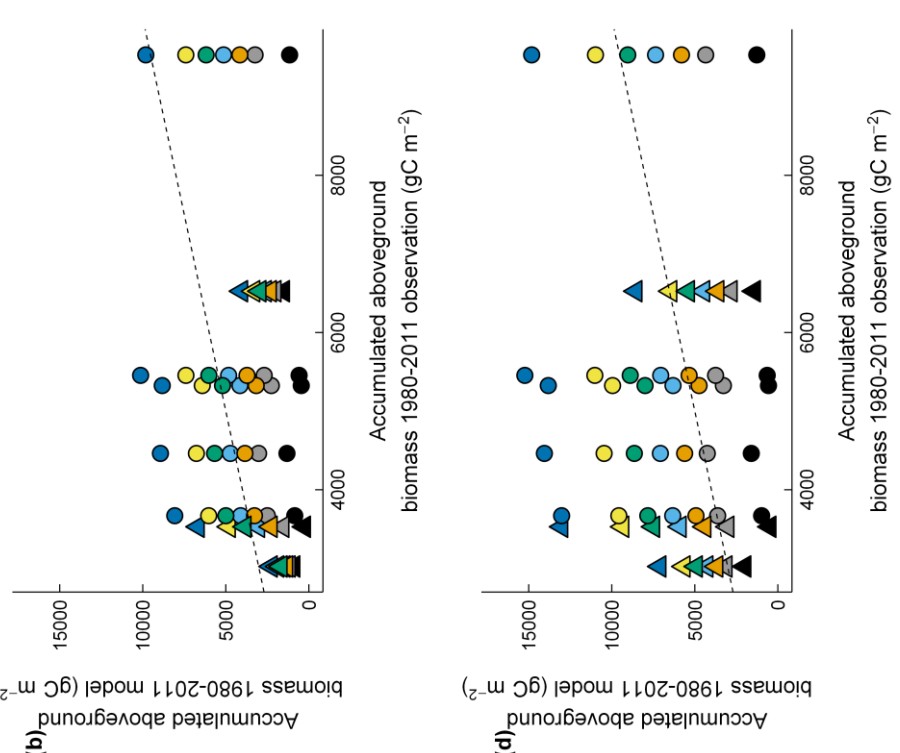

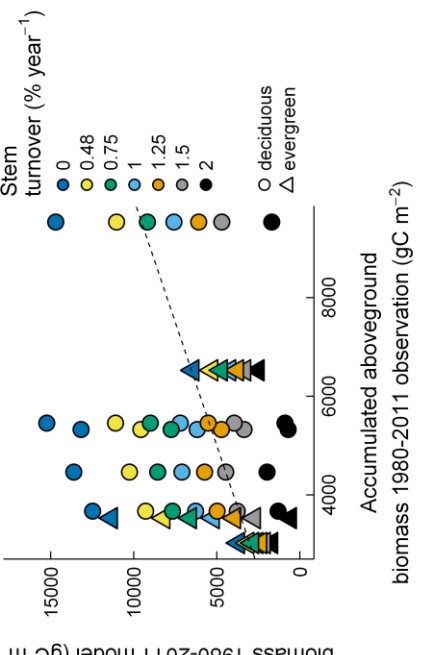

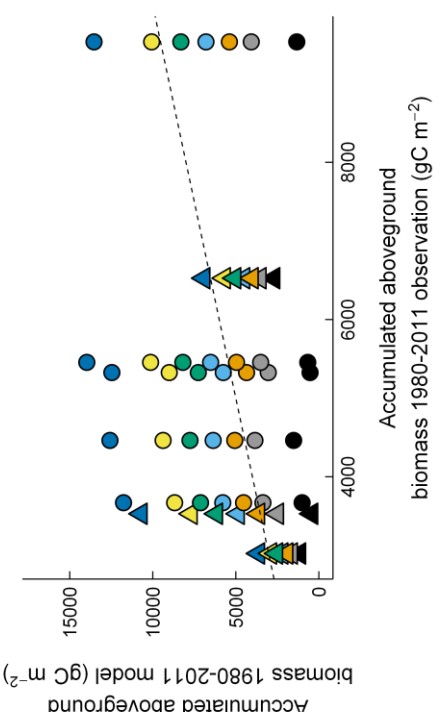





**Figure 7** Comparisons between observed and modeled accumulated aboveground biomass 1980-2011 for (a) D-CLM allocation scheme; (b) D-Litton allocation scheme; (c) F-Evergreen allocation scheme; (d) F-Deciduous allocation scheme. We assumed different turnover rates for stem from 0 to 2% year-1. Turnover rate for stem in the model is 2% year-1. Dashed line is 1:1 relationship between observations and model. Observations (aboveground biomass estimates from tree-ring data) are from Dye et al. (2016) for the Ha1 and Ho1 sites, and following the methodology in Alexander et al. (under review) for the rest of sites.



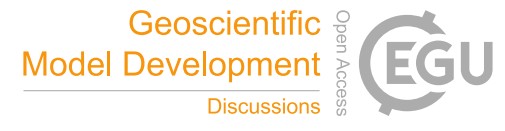

**Figure 8** (a) Comparisons between Cstem/Cleaf ratio for the four C allocation schemes and AmeriFlux observations. (b) NPPstem/NPPleaf ratio for the different mean annual NPP values and C allocation schemes. In 8a, dashed line is 1:1 relationship between observations and model. Observations (Cstem and Cleaf) are from the AmeriFlux database.



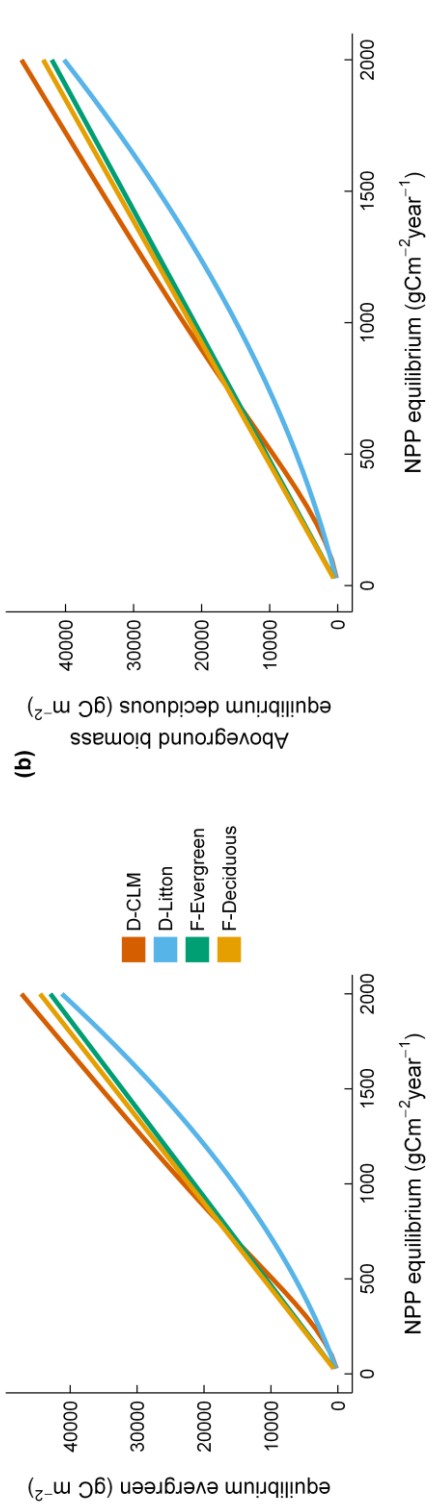

**Figure 9** The C allocation scheme determines aboveground biomass C at equilibrium for (a) evergreen and (b) deciduous sites. For the deciduous sites, with NPP at equilibrium conditions, the D-Litton allocation scheme is closer to the observed aboveground biomass values in 1980 (see Fig. 5a).