# Peer review of "Evaluating the effect of alternative carbon allocation schemes in a land surface model (CLM4.5) on carbon fluxes, pools and turnover in temperate forests"

_Geoscientific Model Development, 2017_

## Referee Comment (RC1) · Anonymous Referee #1 · 18 Apr 2017

This paper assesses the validity of the carbon allocation scheme in the Community Land Model (CLM4.5) against a range of data sets, primarily from flux towers in North America. Two alternative schemes are also introduced as tested against the same data sources.

In general I found this to be a robust piece of model-data comparison, using novel observations and with high relevance to contemporary issues in land surface model development. It also identifies some significant failings of the existing model and suggests schemes that would lead to improvements. I think the organization of the text could be slightly improved upon, but otherwise I have no major concerns here.

**General points:**
Throughout the manuscript, there is a tendency to refer to the CLM4.5 as simply 'CLM'. This should be corrected to, wherever possibly, only using CLM4.5, since all other versions of CLM woud produce substantially different findings to those presented here.

In general, much of the writing in this paper leaves the reader in suspense about the purpose of the paragraph until the end, which makes it hard to follow. I suggest trying to reverse the logic of the paragraph/sections, adding the concluding statements much nearer the beginning, and then backing them up with what follows.

**Specific Points**
P1L34: This 'fixed scheme' is introduced as two seperate schemes, one for evergreen and one for deciduous, but given that (P6L23) these differences are derived only from the model parameters, and thus do not require and structural

modifications, I would see this as one and not two seperate 'schemes'.

P2L10: The last sentence of this paragraph (That coud be done by…) is too vague and should be modified to include specific recommendations.

P2L11-13: This reads as somewhat circular (we need to look at biomass data to get biomass correct)? Maybe remove the very last phrase?

P3L1: Do all LSMs do this? What are the exceptions?

P3L1-17: This long paragraph is somewhat difficult to navigate and could do a better job of clearly introducing the focus of his paper. I suggest splitting it into at least two paragraphs, and making clear what is the focus of this paper early on in each paragraph.

P3L22: What are 'biometric data' in this context?

P4L8: Is there a reference for the L2 product? I wouldn't assume that everyone is 100% familiar with this.

P4L14: Why construct these historical biomass datastreams? Why not just use current biomass estimates? A motivation needs to come before any of this.

P4L24: I'm guessing allometric assumptions have huge impacts on stem carbon estimates and thus on the rest of the results? These should be discussed somewhere.

P5L6: After all the discussion of biomass estimates, more

detail is needed on the leaf C methodologies, since that is non-trivial as well.

P5L9: Not necessary to say PTCLM was used, since this isn't a different model, and makes it seem like it might be.

P5L15: 'dead' stem and root pools are a little confusing. Are these heartwood, or in the litter pool? This needs to be disambiguated.

P5L17: I think all these a1, a2, a3 references should have the numbers in subscript.

P5L30: What is the purpose of fitting these parameters for a broad range of NPP?

P6L24-27: This section seems like results.

P6L30: This sentence make it seem like SLA is prognostic when it is actually fixed.

P7L10: There should be units for these parameters here.

P7L25: What optimization approach? Did you use the adjusted parameters in all of the allocation model simulations, or in a new set of simulations?

P7L27: Is the model emulator 'just' equation 4, or is there more to it that needs explaining?

P7L26: Why are you optimizing turnover? Again, are these new numbers used in all the simulations? There needs to be a better overall narrative connecting the different parts of the

methods section.

P8L5: There needs to be a motivations statement at the beginning of this section, otherwise one has to read all the way through to figure out where the argument is going…

P8L23: These also seem like results.

P9L19: CLM was already introduced much earlier.

P10L6: Why use the CRUNCEP data when these are Ameriflux sites for which meteorological observations are typically available? I don't think this means that the simulations need re-doing, since this paper is focused on relative allocation schemes, but I do think that some more discussion of the potential for errors when comparing site level data and a model driven with reanalysis would be appropriate.

P11L19: This sentence is confusing. State which values - were- used, not which ones were not…

P11L21: I don't understand this 'turnover effect' from this sentence. What are the max and min tem turnover rates?

P12L10: In general this discussion section is rather too long and could do with focusing more coherently on the important findings of the study.

P12L14: What are 'initial' biomass estimates in this context?

P12L16: Couldn't there be different parameters for dec and evergreen plants within the Litton scheme? Why not propose

the ideal scheme within this paper?

P12L28: This sentence is confusing. Why mix the reporting for the evergreen and deciduous forests up like this?

P12L30: Which version of CLM did Hudiberg et al. use?

P13L6: These comparisons with other models are somewhat distracting all the way through this section. I'm not sure it's particularly relevant, given that a) there's no real reason to imagine that there would be a systematic bias and b) the illustrations given are not wide-ranging enough to demonstrate one. I'd suggest moving all that material to its own section or removing it entirely.

P13L5: This seems to suggest that Litton is better but only becasue of the existing biases in NPP?

P13L11: What exactly is being suggested here? I feel like it needs a specific equation.

P14L3: Is the ORCHIDEE bias for the same reason? There are lots of ways to get a high LAI bias!

P14L5L: This root allocation discussion perhaps need to be in its own section (and maybe could be removed since the datasets used here don't really address root allocation per se).

P14L26-: This discussion of 'initial conditions' might also be removed, since 1) this study doesn't really look into initial condition variability 2) it thus doesn't show any sensitivity to initial conditions, 3) what 'initial conditions' are isn't defined

here and 4) the derivation of all of the equilibrium biomass pools earlier terms rather undermines the notion that initial conditions might be important. The IC study used by Kay et al. in particular, illustrates extreme sensitivity to very minor perturbations of atmospheric initialization, but this is not really relevant to the problems presented here.
IC sensitivity is possible in a model like CLM, due to positive feedbacks between low canopy LAI and surface temperature, nitrogen acquisition, etc. but that is not a feature of this analysis.

P15L15: This reference to Xia is confusing. This is with a model other than CLM4.5, but whic one? I'd suggest removing it, since structural modifications of one model are not necessarily relevant to another.

P15L25: What happens at the other sites?

P15L30: This is an important point, maybe highlight more in the abstract, conclusions, etc.

P16L6: This long discussion of Harvard forest rather detracts from why turnover is hard to estimate from the tree ring data? Stem turnover can surely be estimated from permenant sample plot data instead?  Further, this whole discussion is really about how plot level observations (rather than the type of observations) are altered by disturbance history. Big leaf models implicitly aggregate all successional stages together, and so comparison with individual sites is problematic, which is a good argument for using site-specific stem mortality estimates. I think this argument could be made clearer, and shorter!

P16L17: The 'large geographical scales' phrase is repeated from several lines earlier .

P16L18: In CLM, land use change is considered separately from natural ecosystem physiology.

P16L21: Published where? And what numbers were used where? This is too vague of a description.

P17L8: I'm not sure that the allocation schemes disagree, they are just different… Can this be rephrased?

P17L6: The dynamic allocation schemes could be interpreted as plausibly operating at a cohort scale, but also could be interpreted entirely at the landscape scale.  This secton introduces the idea that cohort representation is needed, but then discusses coherent patterns in the site-level stem/leaf ratios, undermining that argument.

P17L14: Better representation of veg dynamic and functional root representation are very different ambitions for LSMs, and have very different implications for allocation schemes. ED-like models, for example, already use allocation schemes that map onto changing stem/leaf ratios with tree size, but these are inappropriate for big leaf models. Some models (LM3-PPA) already have functional roots and change allocation to the accordingly.  It seems like this topic (how to move forward with allocation schemes) is introduced too suddenly in the conclusions, when it might be a better topic for a discussion section evaluating the potential for alternative model improvements to have better connections to data?

---

## Referee Comment (RC2) · Anonymous Referee #2 · 25 Apr 2017

I would only like to add to the comments of the other reviewer the following remarks:

Given that the concept "residence time" is ambiguous (see Sierra 2016 GlobChangBiol, doi: 10.1111/gcb.13556), it would add clarity to the paper if the authors shortly defined the concept.

Some sentences need revision because they are not clear, for example: P2L9-10: in "That could be done..." is not clear if "could" is pointing to events in the past or the future. P3L4-5 P4L12: I would suggest to remove "but not for all" P4L25-6 P8L6: I would suggest to remove "then" P6L17-21

[Figure]

Other comments: The reference of figure 1 in P10L15 does not match the actual figure: (a) NEE (b) GPP (c) Respiration. Also, from the figure it indeed seems like an overestimation of NEE instead of an underestimation.

In P15L20-3, is it possible that the comparisson of predictions of aboveground biomass increment (dynamic) with static observations = rings have caused the observed underestimation? Perhaps the comparison is not valid.

I agree with P16L29-30 in that it should be clear what the turnover rate actually comprises. However, in equation 4 it is clear that $u_i$ is the rate at which material leaves $B_i$, so all C releases (respiration, litterfall, etc.) are necessarilly lumped together in this parameter, unless they are independently specified.

Finally, for Scientific Reproducibility, consider publishing the code in a repository.

---

## Author Comment (AC1) · 15 Jun 2017

June 14, 2017

Dear Editor,

Thank you very much for your letter (gmd-2017-74). Following the reviewers' suggestions we have revised the manuscript which we will upload as we receive instructions.

Below are our detailed replies to the issues raised by the reviewer 1. After each com-

ment we have written a response followed by direct quotes of any changed

The response to reviewer 2 will be completed within one week.

Referee gmd-2017-74-RC1 (Anonymous Referee #1)

This paper assesses the validity of the carbon allocation scheme in the Community Land Model (CLM4.5) against a range of data sets, primarily from flux towers in North America. Two alternative schemes are also introduced as tested against the same data sources.

In general I found this to be a robust piece of model-data comparison, using novel observations and with high relevance to contemporary issues in land surface model development. It also identifies some significant failings of the existing model and suggests schemes that would lead to improvements. I think the organization of the text could be slightly improved upon, but otherwise I have no major concerns here.

We are very pleased that the reviewer finds this work to be robust, relevant and useful in highlighting some new issues in model development. We have attempted to re-organize the text as per the reviewers recommendations.

General points; Throughout the manuscript, there is a tendency to refer to the CLM4.5 as simply 'CLM'. This should be corrected to, wherever possibly, only using CLM4.5, since all other versions of CLM woud produce substantially different findings to those presented here.

We have replaced "CLM" with "CLM4.5" throughout the manuscript. We have modified the legends of the Figures to use "CLM4.5" instead of simply "CLM".

In general, much of the writing in this paper leaves the reader in suspense about the purpose of the paragraph until the end, which makes it hard to follow. I suggest trying to reverse the logic of the paragraph/sections, adding the purpose of the argument much nearer the beginning, and then backing up with the text that follows.

We have revised much of the manuscript for clarity. We have reviewed the opening sentences of each paragraph in an attempt to make the topic clearer. The details of the changes are captured in specific comments below. For example, one major change is the introduction of a paragraph at the start of the methods that provides a guide to the reader.

Specific Points P1L34: The 'fixed scheme' is introduced as two seperate schemes, one for evergreen and one for deciduous, but given that (P6L23) these differences are derived only from the model parameters, and thus do not require and structural modifications, I would see this as one and not two seperate 'schemes'.

We used the word 'scheme' in the hopes of avoiding confusion with 'structure' and to avoid saying "different structures and parmeterizations" throughout the manuscript. We acknowledge that our schemes are not all structurally distinct and we have altered the phrasing to make clear that different allocation schemes arise from different structures and parameterization. For example: Page 3: "...to study the effects of alternative C allocation structures and parameterizations. . ." Section 2.4 is now titled "2.4 Alternative C allocation structures and parameterizations"

In section 2.4 we now introduce the fixed "schemes" as follows: "In addition to the dynamic C allocation structure in CLM4.5 (Oleson et al., 2013), we implemented an alternative dynamic (Litton et al., 2007), and two fixed (Luyssaert et al., 2007) C allocation parameterizations with the same structure." And just below... "The two alternative fixed C schemes have the same structure but different allocation parameterizations and were based on observed values reported by Luyssaert et al. (2007), which were converted accordingly to the allometric parameters used in CLM. One of the C allocation parameter sets was representative of temperate evergreen forests (named "F-Evergreen") and the other of temperate broadleaf deciduous forests (named "F-Deciduous")."

Also in the description of model experiments 2.6 "Each experiment represents a different allocation scheme. For experiment 1 we used the original dynamic C allocation
structure in CLM4.5 (D-CLM4.5; see Sect. 2.3). For experiment 2, we used the alternative dynamic C allocation structure based on Litton et al. (2007) (D-Litton, see Sect. 2.4). For experiments 3 and 4, we used a fixed C allocation structure representative of evergreen (F-Evergreen) and deciduous (F-Deciduous) forests, respectively (Luyssaert et al. 2007 – see Sect. 2.4)."

We think that this avoids confusion and respects the conventional use of the terms "structure" and "parameterization".

P2L10: The last sentence of this paragraph (That coud be done by: : :) is too vague and should be modified to include specific recommendations.

We have modified it and the end of this paragraph now reads: "We identified key structural and parameterization deficits that need refinement to improve the accuracy of LSMs in the near future. These include changing how C is allocated in fixed and dynamic schemes based on data from current forest syntheses and different parameterization of allocation schemes for different forest types."

P2L11-13: This reads as somewhat circular (we need to look at biomass data to get biomass correct)? Maybe remove the very last phrase?

We have modified this and the section now reads: "Our results highlight the utility of using measurements of aboveground biomass to evaluate and constrain the C allocation scheme in LSMs, and also the need for empirical estimates of C turnover rate. Revising these will be critical to improving long-term C processes in LSMs."

P3L1: Do all LSMs do this? What are the exceptions?

No, not all LSMs do this - we re-wrote this section based on the recommendation below. Please see our response to comment P3L1-17

We opted not to include a list, since it would be complicated by lots of variants and option in models - here is a truncated list based on the papers we cite. Terrestrial biosphere models with fixed schemes: CLM, MEL, ED, IBIS, CASA, JSBACH, Triffid,

CABLE, EALCO, GDAY Terrestrial biosphere models with functional relationships or resource dependent schemes: Orchidee variants, CASA variants , LPJ Guess variants, ISAM, TECO, SDGVM For changes in response to this reviewer comment please see the response below to P3L1-17

P3L1-17: This long paragraph is somewhat difficult to navigate and could do a better job of clearly introducing the focus of his paper. I suggest splitting it into at least two paragraphs, and making clear what is the focus of this paper early on in each paragraph.

We rewrote this section in two paragraphs. We believe it now better introduces the study. It now reads as follows:

"Allocation of C between pools in terrestrial ecosystems is poorly represented in LSMs (Delbart et al., 2010; Malhi et al., 2011; Negron-Juarez et al., 2015). Some LSMs use fixed ratios for each Plant Functional Type (PFT), while other models use allocation fractions that are altered by environmental conditions (Wolf et al. 2011; DeKauwe et al 2014). Though many LSMs use the same fractional allocation for both evergreen and deciduous forests, global syntheses show differences in inferred C allocation patterns, for example, the percentage of NPP allocated to leaves that is greater in deciduous than in evergreen forests (Luyssaert et al., 2007). In part this is because it is difficult to measure allocation to different pools at ecosystem or landscape scales and instead we infer what partitioning was required to result in different biomass pools. While eddy covariance observations can be used to parameterize and benchmark LSMs either at single sites or, using geospatial scaling methods, across regions or the globe (Baldocchi et al., 2001; Friend et al., 2007; Randerson et al., 2009; Zaehle and Friend 2010; Mahecha et al., 2010; Bonan et al., 2011), these data inform fluxes in and out but do not provide information on allocation between pools (Richardson et al., 2010).

Studies focusing simultaneously on C pools, fluxes and allocation are relatively rare (Wolf et al., 2011; Xia et al., 2015; Bloom et al., 2016; Thum et al., 2017), in part

because collecting biometric data in addition to flux data is very labour intensive. Some forest inventory data includes estimates of the average biomass within the leaf, wood and root pool, and these can be used to parameterize and benchmark models (Caspersen et al., 2000; Brown, 2002; Houghton, 2005; Keith et al., 2009). The AmeriFlux network provides a rare opportunity to investigate forest allocation processes because gross primary productivity and respiration fluxes are quantified continuously. However, measurements of pool sizes in leaves, stems etc. are less available at these sites and so have been less frequently explored."

P3L22: What are 'biometric data' in this context?

We clarify that biometric data in this context refers to aboveground biomass and leaf area index. It now reads as follows: "We collated biometric data (aboveground biomass and leaf area index), where available, for AmeriFlux sites and supplemented these data with novel aboveground biomass estimates from tree-ring data for AmeriFlux sites (Alexander et al., in review)."

P4L8: Is there a reference for the L2 product? I wouldn't assume that everyone is 100% familiar with this.

Yes, the reference is as follows: Boden, T. A., Krassovski, M., & Yang, B. The AmeriFlux data activity and data system: an evolving collection of data management techniques, tools, products and services. Geoscientific Instrumentation, Methods and Data Systems, 2(1), 165-176, 2013 A pdf of this paper is available here: http://www.geosci-instrum-method-data-syst.net/2/165/2013/gi-2-165-2013.pdf

We have included the reference and website as follows: To quantify carbon flux into and out of the different forests, eddy covariance measurements were collated from the AmeriFlux L2 gap-filled data product (Boden et al. 2013, http://ameriflux.lbl.gov/data/download-data/) for all sites, except for Niwot Ridge where only the AmeriFlux L2 with-gaps data product was available and there we used the REddyProc package (Reichstein et al., 2005) to gap-fill and partition the data (Table

1).

P4L14: Why construct these historical biomass datastreams? Why not just use current biomass estimates? A motivation needs to come before any of this.

Biomass observations are not available from the site PIs for all sites, so we collected dbh and tree ring increments for all the sites. Furthermore growth increment rather than static surveys can be informative for evaluating dynamic allocation schemes. We have added the following text: "To quantify aboveground biomass at all of the sites, we surveyed each forest between 2012 and 2014 and calculated above-ground biomass between 1980 and 2011 (Table 1) using a dendrochronological sampling technique (Dye et al., 2016; Alexander et al., in review). This provided a reconstruction of year-to-year variability in diameter at breast height (dbh) of trees and biomass inferred from allometric regressions. "

P4L24: I'm guessing allometric assumptions have huge impacts on stem carbon estimates and thus on the rest of the results? These should be discussed somewhere.

Theses assumptions are discussed at length in Dye et al. (2016) and Alexander et al. (2017) and in a forthcoming analysis on data assimilation with CLM. Here we include: "At Harvard and Howland, tree-ring reconstructed biomass was compared to biomass estimated from permanent plots established in 1969 and 1989 respectively; tree-ring biomass increment estimates fell within the 95% confidence intervals of biomass estimated from 30 the permanent plots (Dye et al., 2016). Both permanent plots and tree-ring reconstructed biomass are dependent on allometric equations which contributes to uncertainty in these values."

P5L6: After all the discussion of biomass estimates, more detail is needed on the leaf C methodologies, since that is non-trivial as well.

NEE, LAI and a small number of above-ground biomass estimates are available from the AmeriFlux network and so we reference the protocols and sources for these data.

The tree ring based estimates of biomass are novel and so we explain their generation.

We have rewritten this section to make this origin of these data clear:

To quantify the how much carbon was stored in aboveground woody biomass and leaf biomass in these forests we collated already existing biomass and LAI estimates from the AmeriFlux network; these were available for only some sites and years (Table 1). In-situ measured LAI was available from AmeriFlux data for some sites (Table 1), and we used the annual maximum LAI for all the available measurements in each year. We used leaf C-LAI ratio from the AmeriFlux sites with simultaneous measurements of LAI and leaf C during the same year (Table 1). The Cstem/Cleaf ratio, which was derived from AmeriFlux data with Cstem and Cleaf estimates for the same year, was only available for a subset of sites and years (Table 1).

P5L9: Not necessary to say PTCLM was used, since this isn't a different model, and makes it seem like it might be.

We have removed the mention of PTCLM

P5L15: 'dead' stem and root pools are a little confusing. Are these heartwood, or in the litter pool? This needs to be disambiguated.

We do not refer to litter pools, only to leaf, stem and root pools. We refer to Oleson et al. 2013 to help clarify these concepts "CLM4.5 includes the following plant tissue types: leaf, stem (live and dead stem), coarse root (live and dead coarse root), and fine root (Oleson et al., 2013)."

P5L17: I think all these a1, a2, a3 references should have the numbers in subscript.

a1, a2, and a3 references have the numbers in subscript as in Oleson et al. (2013)

P5L30: What is the purpose of fitting these parameters for a broad range of NPP?

In the two dynamic schemes, allocation varies with respect to NPP. We thought it prudent to illustrate this effect so that readers could see the effect of low and high NPP on

the paramters. We converted the allometric parameters to allocation coefficients because these are easier to interpret than the parameters. We have rephrased to make this year. "To account for the range of NPP found in temperate forests, we calculated the allometric parameters a1, a2 and a3 for a broad range of NPP, and then converted the allometric parameters to allocation coefficients for each plant tissue using the C allometry in the model (Oleson et al., 2013). We illustrate in one figure the effect of annual NPP on C allocation to each plant tissue in D-CLM4.5 (Fig. S1)."

P6L24-27:This section seems like results.

The text has been moved to section 3.3, in the Results section.

P6L30: This sentence make it seem like SLA is prognostic when it is actually fixed.

We have now clarified that SLA is a critical and fixed parameter: "CLM4.5 uses a prognostic canopy model, with feedbacks between GPP and LAI acting through allocation to leaf C and SLA and with SLA being a critical fixed parameter in this feedback pathway (Thornton and Zimmermann, 2007)."

P7L10: There should be units for these parameters here.

We have included the units for parameters m and SLA0

P7L25: What optimization approach? Did you use the adjusted parameters in all of the allocation model simulations, or in a new set of simulations?

We clarified that the optimization approach was based on least squares. We used the optimized parameters for temperate forests in all of the allocation schemes simulations (D-CLM4.5, D-Litton, F-Deciduous, and F-Evergreen). We used the default parameter values for m and SLA0 for evergreen forests in all of the allocation schemes simulations.

"and used an optimization approach based on least squares that combined the range of parameter values and Eq. (3) to find the best combination of values for the two

parameters given the leaf C-LAI observations at our sites." And "After optimizing the parameters m and SLA0, we used m=0.0010 and SLA0=0.024 for deciduous forests. For evergreen sites, we could not optimize the parameters m and SLA0 due to the limited number of leaf C-LAI observations available. All model experiments were carried out after SLA optimization."

P7L27: Is the model emulator 'just' equation 4, or is there more to it that needs explaining?

We have clarified that the model emulator was equation 4 (and altered the start of the section to clarify purpose. "We estimated a range of plausible, site specific stem turnover rates using equation (4) below because, at individual research forest stands, rates of tree mortality may or may not reflect averages rates across larger areas". And just below... "To optimize the stem turnover rate we used equation (4) as a model emulator to modify the default stem turnover rate (2%) to within a range of 0 to 2% (van Mantgem et al., 2009; Brown and Schroeder, 1999); for the rest of plant pools we used the default turnover rate in the model."

P7L26: Why are you optimizing turnover? Again, are these new numbers used in all the simulations? There needs to be a better overall narrative connecting the different parts of the methods section.

We have added a section directly under "METHODS" which serves to explain the overall methods narrative. "We implemented the CLM4.5 model – a well-established and commonly used LSM, at nine different forest sites (2.1) and compiled observation of C fluxes, C pools, LAI, and the Cstem/Cleaf ratio (2.2) to evaluate alternative C allocation structures and parameterizations (2.3 & 2.4). We re-parameterized the Specific Leaf Area (SLA) based on available observations (2.5) before implementing four CLM4.5 model experiments designed to examine the impact of the different C allocation structures and parameterizations (2.6). Finally to investigate the potential effects of site variation in woody turnover we estimated plausible site-specific turnover rates (2.7)."

We have re-ordered the methods to make the distinction between CLM4.5 simulations and model emulator studies more explicit. The newly numbered section "2.7 Sensitivity of biomass increment to stem turnover rate" contains the following text to explain the purpose of this analysis:

"We estimated a range of plausible, site specific stem turnover rates using equation (4) below because, at individual research forest stands, rates of tree mortality may or may not reflect averages rates across larger areas. LSMs typically are run at scales that are coarser than individual forest sites and use aggregate estimates for C pool turnover.".

P8L5: There needs to be a motivation statement at the beginning of this section, otherwise one has to read all the way through to figure out where the argument is going. . .

This section may be useful to some readers in understanding the effects of changes in allocation on the aboveground biomass at the start of the runs. However it's length and complexity detract from the main message of the paper and so we have opted to move it to the supplemental material.

The aboveground biomass in equilibrium conditions will depend on aboveground NPP (ANPP), the NPPstem/NPPleaf ratio (or astem/aleaf ratio) and the turnover rates for leaf and stem (uleaf and ustem).

P8L23: These also seem like results.

This section may be useful to some readers in understanding the effects of changes in allocation on the aboveground biomass at the start of the runs. However it's length and complexity detract from the main message of the paper and so we have opted to move it to the supplemental material.

P9L19: CLM was already introduced much earlier.

This reference to CLM has been removed

P10L6: Why use the CRUNCEP data when these are Ameriflux sites for which meteorological observations are typically available? I don't think this means that the simulations need re-doing, since this paper is focused on relative allocation schemes, but I do think that some more discussion of the potential for errors when comparing site level data and a model driven with reanalysis would be appropriate.

AmeriFlux sites extend only a decade or so, but changes in biomass are slow relative to ecosystem exchange. To explore the results of slowly changing processes we extended model runs to 30 years which requires using CRUNCEP or some other reanalysis climate. We have added an explanation to this section

"The standard climate forcing provided with the model is the 1901-2013 CRUNCEP dataset. While meteorological data is available at the AmeriFlux sites, this data extends only as long at the eddy covariance observations which is less than a decade in several cases. To explore the effects of allocation on slowly changing C pools like woody biomass, we extended model runs to 30 years which requires using CRUNCEP or some other reanalysis climate. "

P11L19: This sentence is confusing. State which values -were- used, not which ones were not. . .

Comments P11L19 and P11L21 are addressed together (see below)

P11L21: I don't understand this 'turnover effect' from this sentence. What are the max and min tem turnover rates?

We have re-written this entire paragraph as follows: "The stem turnover rate that best matched the biomass accumulation rate estimated from the tree ring reconstructions varied by site and was always lower than the default rate of 2% year-1 used in CLM4.5 (Fig 7). As expected, changing the turnover rate had the largest influence at sites with the highest average NPP. Biomass accumulation in the D-Litton scheme was less sensitive to changes in turnover rate compared to the D-CLM scheme (compare Fig 7b to 7a). The F-Deciduous and F-Evergreen parameterization were similar in their

sensitivity to changes in turnover rate (compare Fig 7c to 7d)."

P12L10: In general this discusión section is rather too long and could do with focusing more coherently on the important findings of the study.

We have re-organized and shorted specific sections to clarify the most important findings. See in particular the response to "P15L30: This is an important point, maybe highlight more in the abstract, conclusions, etc."

P12L14: What are 'initial' biomass estimates in this context?

We have rephrased and referred to the relevant figure (Fig 5) to clarify meaning and context. "....it produces lower, and more credible aboveground biomass estimates at the start of the simulation for these forests (Fig. 5a) and matches the biometric estimates of C partitioning between leaf and stem (Fig. 8a)."

P12L16: Couldn't there be different parameters for dec and evergreen plants within the Litton scheme? Why not propose the ideal scheme within this paper?

The relationships in Litton et al (2007), we used to derive the D-Litton structure and parameters were for all forest types. Litton et al (2007) were unable to find differences between forest types in these relationships and so we are reluctant to suggest differential parameterization of this model structure without potential parameters.

P12L28: This sentence is confusing. Why mix the reporting for the evergreen and deciduous forests up like this?

We agree - we have clarified as follows: "None of the allocation schemes simultaneously matched observed evergreen and deciduous forest aboveground biomass."

P12L30: Which version of CLM did Hudiberg et al. use?

Hudiburg et al. 2013 used CLM4 . We have rewritten this sentence as follows: "These results are in line with previous findings in evergreen Oregon forests where CLM4.0 also underestimated aboveground biomass at most sites (Hudiburg et al., 2013)."

P13L6: These comparisons with other models are somewhat distracting all the way through this section. I'm not sure it's particularly relevant, given that a) there's no real reason to imagine that there would be a systematic bias and b) the illustrations given are not wide-ranging enough to demonstrate one. I'd suggest moving all that material to its own section or removing it entirely.

Allocation schemes among these different models are relatively simple and quite similar in structure. Also while the sites we evaluated are not comprehensive, the resources used to parameterize the schemes are quite wide ranging (Litton et al., 2007; Luyssaert et al., 2007) - we think that it is useful to include Table 3, but we agree that there is some confusion here. We have re-organized this section and gathered comparisons into section "4.2 C allocation scheme: implications for C pools"

P13L5: This seems to suggest that Litton is better but only because of the existing biases in NPP?

We cannot exclude this possibility. We try to evaluate whether NPP or stem turnover is overestimated in section 4.5. But note that Fig 8 suggests that the Litton is also better at predicting partitioning between stem and leaf C on average.

P13L11: What exactly is being suggested here? I feel like it needs a specific equation.

We have rephrased for clarity: "Our results suggest that it is necessary to improve the D-CLM4.5 scheme for temperate forests; for evergreen forests the D-Litton scheme could be modified from a linear to a non-linear scheme to increase allocation to stem for sites with mean annual NPP<500 g Cm-2year-1."

P14L3: Is the ORCHIDEE bias for the same reason? There are lots of ways to get a high LAI bias!

PENDING The ORCHIDEE model did not have the identical problem. The point we were addressing here is strong and obvious bias/model inconsistency should be dealt with before you evaluate model changes. To clarify this point, we have re-phrased

as follows: "Clear and persistent model-data discrepancies in LAI also needed to be addressed in the ORCHIDEE LSM prior to any evaluation of model changes (Thum et al., 2017). Site specific estimates of SLA and LAI would be very useful for optimizing parameters within their observed range and allow mechanistic processes controlling allocation to leaves in the model to be assessed."

P14L5L: This root allocation discussion perhaps need to be in its own section (and maybe could be removed since the datasets used here don't really address root allocation per se).

We do not wish to remove this section because any change in allocation to stem and leaf must have an impact on allocation to roots. This section has been renumbered "4.2 C allocation scheme: implications for C pools" and we think that roots should be included here as one of three major C pools we discuss in the paper.We think that it We have shortened this paragraph and included a reference to Weng et al., 2015 in response to the reviewers later comment on optimization.

P14L26-: This discussion of 'initial conditions' might also be removed, since 1) this study doesn't really look into initial condition variability 2) it thus doesn't show any sensitivity to initial conditions, 3) what 'initial conditions' are isn't defined here and 4) the derivation of all of the equilibrium biomass pools earlier terms rather undermines the notion that initial conditions might be important. The IC study used by Kay et al. in particular, illustrates extreme sensitivity to very minor perturbations of atmospheric initialization, but this is not really relevant to the problems presented here. IC sensitivity is possible in a model like CLM, due to positive feedbacks between low canopy LAI and surface temperature, nitrogen acquisition, etc. but that is not a feature of this analysis.

Our manuscript deals only with the land surface portion of the model. The biomass at time 0 will strongly influence biomass at any time in the future. The different allocation schemes result is different biomass estimates (leaf and woody) and since the C fluxes are in some measure proportional to pool sizes we think it is important to discuss this.

We mean to refer to the value of the various C pools at the start of the evaluation runs (1980). We have clarified to avoid confusion with initial atmospheric conditions in studies like Kay et al.

"There is an increasing awareness in Earth system modeling of the critical role of initial conditions (including the initial size of C pools - examined in this study) that adds an extra layer of complexity in diagnosing the impact of an incorrect representation of physical processes on the transient simulation (Kay et al., 2015; Fisher et al., 2015)."

P15L15: This referente to Xia is confusing. This is with a model other than CLM4.5, but which one? I'd suggest removing it, since structural modifications of one model are not necessarily relevant to another.

We have clarified the reference to Xia and explained 1) that the allocation scheme is very similar to the fixed schemes presented in our study and 2) that when challenged with data they come to the same conclusion as we do. We also found important overestimations of aboveground biomass for deciduous forests with D-CLM4.5, and therefore suggest that the NPPstem/NPPleaf ratio in the model is one of the primary factors contributing to these overestimations of biomass. Overestimation of allocation to stem was also found using the IBIS model, where a fixed allocation scheme with terms for allocation to leaf, stem and root, which sum to 1, was found to overestimate allocation to stem (Xia et al., 2015). The fractional allocation to stem in IBIS was reduced from 0.5 to 0.36 when the scheme was optimized against satellite LAI observations (Xia et al., 2015). Similarly, our results suggest that allocation to stem in D-CLM4.5 should decrease, whereas allocation to leaf and root should increase, in order to align simulated and observed biomass.

P15L25: What happens at the other sites?

Unfortunately, we have no observations from the other sites.

P15L30: This is an important point, maybe highlight more in the abstract, conclusions,

etc.

We have been more explicit in this finding in the abstract "Our results highlight the utility of using measurements of aboveground biomass to evaluate and constrain the C allocation scheme in LSMs, and suggest that stem turnover is overestimated by CLM4.5 for these Ameriflux forests." ...and 4.5 Conclusions and Perspectives we state: "Finally, we show that information on stem turnover rate, which varies with forest age and successional status, is important to interpret the success or failure of different model schemes at forest sites. Default stem turnover in CLM4.5 may approximate steady state conditions at large scales but it is inconsistent with forests which are not at steady state. Decreasing the stem turnover rate from 2% yr-1 to plausible values consistent with their successional status yielded aboveground biomass accumulation rates more consistent with observations."

Also see the response to comment P16L6

P16L6: This long discussion of Harvard forest rather detracts from why turnover is hard to estimate from the tree ring data? Stem turnover can surely be estimated from permenant sample plot data instead? Further, this whole discussion is really about how plot level observations (rather than the type of observations) are altered by disturbance history. Big leaf models implicitly aggregate all successional stages together, and so comparison with individual sites is problematic, which is a good argument for using site-specific stem mortality estimates. I think this argument could be made clearer, and shorter!

This section has been re-organized in an attempt highlight that NPP and turnover are likely overestimated as the reviewer asked in comment P15L30. It has also been shortened by 200 words

The paragraph now reads: "It is likely that CLM4.5 overestimates stem turnover at these sites. Currently, CLM4.5 assumes a stem mortality rate of 2% yr-1 that is higher than published tree mortality rates for forests in the USA (van Mantgem et al.,

2009; Brown and Schroeder, 1999; Runkle, 1998). When considering large geographic scales the 2% yr-1 rate of stem turnover may be reasonable but at individual sites this may be a poor approximation. The Harvard Forest, for example, is at the end of the stem exclusions stage of forest development and, there has been little to no canopy disturbance since the time of the 1969 census. As such, the tree-ring biomass increment estimates at Harvard assume zero mortality between 1980 and 2012. This assumption appears solid as it results in no significant difference between tree-ring reconstructed biomass increment and the repeated measurements from permanent plots over the last 40 years (Dye et al., 2016). We thus decreased stem mortality rate from 2% yr-1 to published ranges of tree mortality (between 0 and 1.5% yr-1), to estimate plausible stem turnover rates for each site and scheme. The resulting ranges of aboveground biomass increment overlapped with the observed aboveground biomass increment estimated from tree-ring data, for nearly all the carbon allocation schemes (see Fig. 7). For Harvard forest the turnover rate that most consistent with the tree ring reconstruction was never zero, which indicates that both NPP and turnover are overestimated for this site in all the allocation schemes. A different turnover rate was required for each site and C allocation scheme to match the observed aboveground biomass increment but in each case it was below the default 2% value. Our analysis suggests that when using AmeriFlux sites to inform models, or other site level observations, taking note of site specific rates of stem turnover is prudent."

P16L17: The 'large geographical scales' phrase is repeated from several lines earlier .

The second instance has been removed in the revision

P16L18: In CLM, land use change is considered separately from natural ecosystem physiology.

In this sentence we were referring to the "real" ecosystem C turnover rate, which will encompass all the things that are mentioned, including LC change. However the reference has been removed in the revised paragraph in response to P16L6.

[Figure]

P16L21: Published where? And what numbers were used where? This is too vague of a description.

The references are now included. They are van Mantgem et al., 2009; Brown and Schroeder, 1999; Runkle, 1998 - see the revised paragraph quoted in response to P16L6

P17L8: I'm not sure that the allocation schemes disagree, they are just different…Can this be rephrased?

We have used the word differ instead.

P17L6: The dynamic allocation schemes could be interpreted as plausibly operating at a cohort scale, but also could be interpreted entirely at the landscape scale. This secton introduces the idea that cohort representation is needed, but then discusses coherent patterns in the site-level stem/leaf ratios, underminng that argument.

Thanks for pointing this out - that was the not the emphasis we were trying to convey. We mean to suggest that Cohort representation could remove the conflation of ontogeny (forest stands change allocation as they mature) with resource optimization (trees allocate resources to maximize growth) that is inherent in the D-CLM scheme. We agree that the order of paragraphs is not helpful. We have moved this paragraph to the end of the paper.

P17L14: Better representation of veg dynamic and functional root representation are very different ambitions for LSMs, and have very different implications for allocation schemes. ED-like models, for example, already use allocation schemes that map onto changing stem/leaf ratios with tree size, but these are inappropriate for big leaf models. Some models (LM3-PPA) already have functional roots and change allocation to the acoordingly. It seems like this topic (how to move forward with allocation schemes) is introduced too suddenly in the conclusions, when it might be a better topic for a discussion section evaluating the potential for alternative model improvements to have

better connections to data?

We agree that these two developments have different implications for allocation schemes, this is why they must be considered together in the perspectives section. While different dynamic allocation schemes might "work" for a given model configuration, none of the current working schemes appear to represent the actual ecological process. We have changed the topic sentence moved this perspective to the end of the conclusions and perspectives section as a 'closing thought'. The line of reasoning is introduced earlier in the discussion -re: root function - at the end of 4.2 "However, some studies suggest that this trade-off includes fine roots (Wolf et al., 2011; Malhi et al., 2011; Chen et al., 2013), probably due to the co-limitation of productivity by resources captured aboveground (e.g. light) and belowground (e.g. nutrients and water) (Dybzinski et al., 2011; Weng et al., 2015). These growth drivers also vary with time and across spatial ecological gradients (Guillemot et al., 2015). In CLM4.5 employed here, the roots control water uptake but are not related to nutrient uptake which limits the potential for dynamic responses to nutrients and CO2 concentrations (Atkin, 2016 De Kauwe et al., 2014; Hickler et al., 2015; Sevanto and Dickman 2015). Root functionality in LSMs could be enhanced by improving parameterization within models and introducing new components such as dynamic root distribution and root functional traits linked to resource extraction (Warren et al., 2015; Brzostek et al., 2014; Shi et al., 2016; Phillips et al., 2016; Brzostek et al., 2017; Iversen et al., 2017)."

Note - we have now included the reference to Weng et al 2015 in that section.

Re cohorts - we have re-written the paragraph as follows: "Ecological theory suggests that dynamic allocation probably reflects whatever resource is most limiting but developing allocation schemes for LSMs that respond to resource limitation is challenging. The two dynamic allocation schemes reflect forest stand development to some extent i.e. as trees get bigger (and can grow more) they tend to invest more in stem and less in leaves. However the two schemes also use low NPP as a proxy for resource limitation, but they differ on how allocation changes as a function of NPP (Fig. S1).

[Figure]

This is a problem because these dynamic schemes cause sites that have low NPP to perpetually allocate more resources to leaves and roots while sites with high NPP perpetually allocate less resources to leaves and roots (Fig. S1). Cohort representation in the model could help deal with this problem by treating allocation caused by low resources differently from early development. As coupled C-N and functional root subroutines are developed for LSMs (Shi et al., 2016), and with better representation of vegetation dynamics (Fisher et al., 2015), we could imagine a dynamic allocation scheme for CLM4.5 based on whether above ground (light) or below ground (water and nutrients) are limiting."

---

## Author Response (AR2)

Dear Editor,

Thank you very much for your letter (gmd-2017-74). Following the reviewers' suggestions we have revised the manuscript. Below are our detailed replies to the issues raised

5    *1. I found confusing your answer to reviewer 1 comment on initial conditions and the corresponding section 4.3. I think there is confusing between the initial conditions of the model run and the spin-up. In both cases you need to start your C stocks from some value, which can be defined as initial condition. I have the impression that Reviewer 1 was referring to the initial condition previous to the spin-up, and your answer and sec 4.3 focuses on the initial condition after spin-up. This difference is important because as you show with equation 4, this model must always go to the same steady-state value*

10   *independent on the initial condition. This is a mathematical property of all linear autonomous models. However, if you get different steady-state values for different initial conditions, is probably because you have non-linearities that are not captured in eq. 4. This is an important point that requires clarification. You must either confirm that this model behaves as a linear autonomous system expressed as eq. 4, and rewrite your discussion on sec 4.3 mentioning that the model will always reach the same steady-state independent on initial conditions, or you modify equation 4 so the reader better understand*

15   *what type of non linearity is included in the model that leads to different steady-state values for different initial conditions.*

There was a confusion between "initial conditions" and the conditions we used to start our comparisons. We have re-written section 4.3 to clarify what we actually did.

**"4.3 C allocation scheme: implications for steady state aboveground biomass**

20   Initial conditions used to begin transient runs or make forecasts in LSMs are usually obtained by spin-up methods. Starting from bare ground, with prescribed physical soil characteristics and plant functional type fractions a time series of meteorological forcing variables are cycled repeatedly until the model reaches a steady-state, a point when C pool sizes and fluxes remain constant between subsequent meteorological forcing cycles. This feature is exploited by Xia et al. (2012) with their semi-analytical approach to calculating these steady-state conditions. Model simulations over timescales from days to

25   centuries critically depend on the initial variable values obtained after spin-up and flawed initial values may produce model output that can be severely biased or unrealistic (Yang et al., 1995; Cosgrove et al., 2003; Rodell et al., 2005; Li et al., 2009). There is an increasing awareness in Earth system modeling of the critical role of these initial values after spin-up (including the initialized size of C pools - examined in this study) that adds an extra layer of complexity in diagnosing the impact of an incorrect representation of physical processes on the transient simulation (Kay et al., 2015; Fisher et al., 2015). Our results

30   reinforce that concern by showing that with the same climate forcing different C allocation schemes within the same LSM can produce strongly differing initial conditions after spin-up for aboveground biomass (Fig. 9). In the Supplementary Methods and Figures, we provide an explanation for the variability in steady state aboveground biomass depending on the C

allocation scheme used in CLM4.5. In the C allocation schemes used, changing biomass with time can be expressed as Eq. (4), which are models that behave as a linear autonomous system (Sierra et al., 2017). This implies the models, when forced with equivalent meteorology and physical soil properties, will eventually converge to a steady-state independent of the starting values of the state variables, although in the case of CLM this may take many tens of thousands of years."

*2. The repository that you linked to in GitHub doesn't contain the parameter files that you used to run the model as advertised. Please include these parameter files. Also, please consider Zenodo (https://zenodo.org/) for long-term archiving of your supplementary repository.*

10    We have uploaded all the parameter files used and the source code modifications at the GitHub repository that we linked (https://github.com/davidjpmoore/gmd-2017-74). For clarity we renamed the parameter files to have a similar name to the one used in the manuscript, for instance D-CLM4.5_EvergreenSite_parameters.nc, D-CLM4.5_DeciduousSite_parameters.nc, D-Litton_EvergreenSite_parameters.nc, etc. I edited the section "Notes on code modifications" to update the names of parameter files, and clarify the parameter file needed for each C allocation scheme,

15    and if it was necessary to use source code modifications

[revised manuscript text omitted]

---

## Author Response (AR3)

Dear Editor

We have made no substantive changes to the main text of the manuscript but the follow modifications were made.

1) The affiliation of Flurin Babst was revised.

2) The acknowledgements were revised to include individual award numbers that supported the work

*Acknowledgements.*

This study was supported by the DOE Regional and Global Climate Modeling DE-SC0016011, the NSF Macrosystems Award 1241851 and 1241930 and by the University of Arizona Water, Environment, and Energy Solutions (WEES) 578 and Sustainability of Semi-Arid Hydrology and Riparian Areas (SAHRA) programs. The US-NR1, US-UMB, and US-MMS AmeriFlux sites are currently supported by the US DOE, Office of Science, through the AmeriFlux Management Project (AMP) at Lawrence Berkeley National Laboratory under award numbers 7094866, 7096915 and 7068666, respectively. AmeriFlux site US-MOz is supported by the U.S. Department of Energy, Office of Science, Office of Biological and Environmental Research Program, through Oak Ridge National Laboratory's Terrestrial Ecosystem Science (TES) Science Focus Area (SFA). ORNL is managed by UT-Battelle, LLC, for the U.S. DOE under contract DE-AC05-00OR22725. FB acknowledges funding from the Swiss National Science Foundation (#P300P2_154543) and the EU-Horizon 2020 Project "BACI" (#640176).

Best wishes,
David Moore, Francesc Montane
Corresponding authors